# A distributed framework for zero-day malware detection using federated ensemble models

Hassan Ishfaq[1], Jamal Hussain Shah [1]*, Rabia Saleem[2], Maira Afzal[1]

**1** Department of Computer Science, COMSATS University Islamabad, Wah Cantt, Pakistan,
**2** Department of Computer Science, Government College University, Faisalabad, Pakistan

* jhshah@ciitwah.edu.pk

## Abstract

Classification and detection of zero-day attacks remain a significant challenge within the domain of cybersecurity. Due to the vast types of malware families and the presence of an imbalanced dataset, real-time detection and classification become increasingly complex and inaccurate. Thus, there's an urgent need to develop an intelligent and adaptive defense mechanism capable of identifying and classifying such attacks with improved precision and robustness. This paper proposed a stacked ensemble federated learning model with an accuracy-aware node weighting scheme to address the challenges posed by inter- and intra-class similarities among different types of malwares. In the initial phase, malware Portable Executable (PE) files are collected from multiple online repositories and validated by three different antivirus programs through VirusTotal to ensure reliability. These validated files are then converted into image form and categorized into 28 families to facilitate feature extraction. In the second phase, deep feature representations are extracted through a transfer learning-based fine-tuned ResNet-50 model, which captures both low-level and high-level patterns that are relevant to malware classification. After feature extraction from multiple distributed nodes, architecture is fed into the novel proposed Ensemble Stacked Federated Model for enhanced generalization and robust classification. The model is tested on both private and publicly available datasets. The experimental results demonstrate that the proposed method outperforms existing baseline approaches in terms of accuracy and computational efficiency. This improvement is achieved because it performs independent training at each federated node separately and then stacks their outputs with a central ensemble model, which enhances the learning rate and reduces overfitting. The code used for the experiments is available here.

## 1. Introduction

The cybercrimes and cyberattacks including malware have seen an impressive and the consistent rise recently. Even with significant advancements within modern

**Data availability statement:** The dataset supporting the findings of this study is publicly available in the FigShare repository at https://doi.org/10.6084/m9.figshare.29107697, and code at Github (https://github.com/jhshah101/Zero-Day-Malware-Detection) All data are fully anonymized and sufficient to reproduce the results reported in this manuscript.

**Funding:** The author(s) received no specific funding for this work.

**Competing interests:** The authors have declared that no competing interests exist.

techniques of cybersecurity and their continuous development, the malware remains one of the dangerous and persistent threats in the cybersecurity domain [1]. Field of malware analysis consists of methodologies from the various domains, including reverse engineering, program analysis, machine learning, and the network analysis. The aim is to investigate the malicious samples comprehensively, achieve deep behavioral insights, understand the attacks and their patterns, and explore how they adapt and change by the time. Recently, malicious software attacks have appeared as critical security challenges, results in widespread damage and financial losses. In 2017, global ransomware, also called as WannaCry, infected over 230,000 devices across 150 countries, resulting in an estimated $8 billion in losses [2]. Similarly, in March 2019, Norsk Hydro, one of the world's largest aluminum producers in Norway, fell victim to new sophisticated ransomware named LockerGoga, that led to temporary closure of multiple production plants and other severe disruptions [3]. Such incidents shows the growing severity and frequency of the malware attacks, critical security risks to industrial production and the massive financial losses. The gradually increasing number of malware variations and the complexity, an accurate and immediate classification of such evolving threats is an utmost need for maintaining the cybersecurity [4]. The traditional ML (machine learning) methods suffer from different limitations, mostly in manual feature engineering, scalability and handling the imbalanced and large volumes of malware datasets effectively [5]. Therefore, there is a strong need to introduce a new way of handling such limitations. One of such efficient methods is DL (deep-based) malware image based classification [6] that has emerged like effective and robust pattern recognition method.

The recent methods for classifying the malware mainly depend on either dynamic or static analyses in order to extract the representative features [7]. Such techniques include examining the bytecode of malware, file headers, assembly code structure, or execution behavior and their dynamic actions, followed by utilizing the ML (machine learning) algorithms for the classification [8]. However, extracting the static features from malware disassembly code often encounters challenges related with reverse engineering specially when malware is obfuscated. In cases where the static analysis proves ineffective, dynamic analysis can be used to directly observe the malware behaviors in runtime. However, dynamic analysis is considered as limited because of its capability to only examine the particular execution paths. This also acts as the evasion method because of its environmental dependency, that makes it time taking and specially resource-intensive [9]. These approaches not only handle challenges of reverse analysis and time efficiency but also depend on manual feature creation heavily, that reduce scalability and the adaptability. So, keeping pace with rapid creation of malware variants becomes a unachievable and complex task. In order to address these threats successfully, the DL (deep learning) based methods emerged as the effective solution [10]. Basically deep learning is characterized by its multilayer architecture, exhibits the remarkable abilities to learn automatically from both the labeled and unlabeled data, that remove the requirement of handcrafted features. Additionally, most of existing methods directly transform the malware binaries into grayscale images [11], in which each byte represents as a pixel. Such grayscale

images must be standardized to fixed size, that are then utilized for training a classification model. To accomplish this uniformity, methods like image scaling, byte truncation, and zero padding are commonly employed because of variations in malware sizes. However, training a deep model on extensive datasets needs computational resource and significant time which is unrealistic for distributed real-time environments. To address this training challenge, a practical approach involves utilizing pre-trained deep learning network architectures, such as ResNet or VGG, to automatically extract features for constructing the classification model [12]. This approach inspires the development of hybrid deep learning architecture for malware family identification and classification based on pre-trained CNNs. These CNNs allow fast training, better generalization and more efficient and effective threat detection even in federated learning and limited computational environment. Despite that, prior federated learning framework depends on simple ensemble learning, and parameter averaging (FedAvg). These traditional federated learning models are limited to handle class imbalance and data distribution across multiple nodes. To overcome this, an approach is introduced in this research work that uses staked ensemble technique combined with accuracy-aware node weighting to enhance zero-day malware detection.

## 2. Research contribution

The main contribution of this proposed research is to present a robust malware families classification approach that improves accuracy in zero-day attacks and inter and intra-classification using ensemble federated learning framework. The key contributions of the proposed work include the following:

1) A novel stacked ensemble federated model is proposed for robust malware classification. In this framework, features extracted from different distributed nodes are combined and their outputs are fused through a central ensemble. By fusing both node-level and global level prediction, model improves classification performance and generalization as compared to traditional multi-node fusion techniques.

2) Fine-tuned ResNet-50 used as backbone that provides lowest FLOPs and faster inference time to extract features from PE files. These features are then converted into image representation that are utilized within federated pipeline to reduce training cost while preserving classification accuracy.

3) This model is designed to address the high structural similarities challenge that misclassify zero-day attacks by employing federated ensemble stacking architecture.

## 3. Related work

Over the last two decades, there has been significant development in Machine Learning (ML) and Deep Learning (DL) techniques in literature for the detection and classification of malware. Traditional methods for this work are divided into two main categories, static analysis or dynamic analysis to obtain a feature sets [13]. Static malware analysis doesn't require actual execution of code, and delivers benefits in terms of processing speed and computational efficiency. In comparison, dynamic analysis involves automatic execution of malware within a sandbox or controlled environment to observe its behavior. Furthermore, variety of techniques are also involved in analyzing bytecode, disassembly code, and file structure [9]. However, this paper focus is on static malware detection and classification due to its efficiency and scalability.

Many researchers have proposed deep learning techniques-based techniques for static malware detection such as Rajasekhar C. et. al. [14] introduces a CNN model powered by DL for the classification of Portable Executable (PE) binary files. This classification is carried out by utilizing a fusion feature set approach that enhances representation and classification. The performance assessment of proposed model demonstrates its capability to accurately classify malware and benign files, achieving an impressive 97% accuracy when employing fusion feature sets. Similar, Mumtaz A. et al. [15] proposed a hybrid deep and ML approach to represent and analyze the malware signatures within the BIG15 dataset which spans nine distinct classes. Huaxin D. et al [16] also introduced same image based malware classification

 

technique using deep transfer learning based approach to classify malware into nine different classes, achieving an exceptional accuracy of 99.44% on Microsoft's public malware dataset. In [17], Z. Zhao et al. introduced similar image visualized based technique but extracted different deep features to gain the malware accuracy especially on Malimg dataset. They extended the AlexNet transfer learning based for malware detection and classification in which they evaluated their model on two publicly available dataset namely Microsoft malware dataset and the Google Code Jam (GCJ) to validate their model adaptability and generalization. On the other hand, in [18], Stephen O. S. et. al. presents a hybrid framework for malware classification, to address the limitations of existing static and dynamic methods. The framework combines advanced techniques for both static and dynamic malware analyses. Furthermore, it transforms static malware executables and dynamic process memory dumps into images, which are then mapped using space-filling curves to extract visual features for classification. In [19], Tobiyama et al. highlight the increasing limitation of traffic data analysis due to increasing sophistication of the malware that mimics benign network traffic. Proposed method leverages the deep neural networks, specifically the Recurrent Neural Networks (RNNs) and the Convolutional Neural Networks (CNNs), in order to detect malware based on the process behavior. This method involves training an RNN to extract the temporal features from API call sequences, which are used to generate the feature images representation classified by CNN. Dataset used includes process logs from 81 malware and 69 benign processes, collected in the controlled environment using tools like Process Monitor and the INetSim. Evaluation through 5-fold cross-validation demonstrated the high accuracy with best configuration achieving an AUC of 0.96. However, the limitations include dependency on the predefined presence of unseen and potential operations in the validation data, necessitating further research in order to enhance the generalization and the real-world development. In [20] A. F. Agarap developed an intelligent anti-malware system using DL models combined with SVM (Support Vector Machine) classifier in order to identify the unknown malware families. Using the Malimg dataset, that contains 9,339 images of malware across 25 families, study trains 3 deep learning models: GRU-SVM, CNN-SVM, and MLP-SVM. Preprocessing involved resizing the images to matric 32x32 and standardizing features. Between these models, GRU-SVM demonstrated the highest predictive accuracy of 84.92%, attributed to its five layer intelligent architecture. Nevertheless, study also revealed GRU-SVM's longer training time, showing the trade-off among the predictive performance and computational efficiency. CNN-SVM and MLP-SVM accomplished the lower accuracies of around 77.23% and 80.47%, respectively. The study suggests that maximizing the architectural complexity of CNN-SVM and MLP-SVM could boost the performance. Limitations include computational demands of more complicated models and potential need for more diverse datasets to generalize the findings across different malware families.

In [21], different methods and results regarding malware classification are explored. This study employs novel method through converting malware samples into the grayscale images and extracting both global and local image features in order to train the ML classifiers. The methods include the use of 5 datasets, each containing 4850 samples, and training the 6 different classifiers: K-Nearest Neighbour, Random Forest, Bagging, AdaBoost, Decision Tree, and Gradient Boost. Local features like SURF, ORB, SIFT and KAZE, are utilized with the global features such as Colour Histogram, Hu Moments and Haralick Texture. The research demonstrates the hybridizing classifiers on the basis of these features result in high classification accuracy of up to 98.2%. In addition, DL models ResNet and Inception V3 were tested, accomplishing over 96% accuracy in some experiments. Limitations include the potential vulnerabilities in code obfuscation and manipulation methods, that affect the static analysis methodologies.

A multiple DL methodologies on malware classification are analyzed in [22] that utilizes the CNN and integrates it with the SVM (CNN-SVM) in order to classify malware families. This study also utilizes Malimg dataset which comprises of 9,339 samples from 25 distinct malware families. CNN model accomplished highest test accuracy of 97.5%, outperforming the other models. Key methods involve converting the malware binaries to the grayscale images and using GIST descriptors for the texture features, with k-nearest neighbors for classification. The research also reviews the performance of different classifiers, noting CNN's higher accuracy. Limitations mentioned include the requirement for the improved architectural designs in DL models and the challenge of effectively processing the large signature databases for IoT

devices. A study in [23] presents a malware variant detection methodology in IoT (Internet of Things) environment. Traditional malware detection methods struggle to adapt to IoT platforms because of the significant differences in terminals and OS (operating systems). Proposed methodology addresses this issue by introducing a feature representation method that is based on RGB images, concentrating on developer metadata and assembly code of malware. This method enriches the texture information in order to establish deeper links among the IoT variants and the original malicious code. Moreover, this methodology maximizes CNN model by integrating self-attention mechanism and the spatial pyramid pooling to meet the varying image dimensions of an IoT malware samples. Experimental results confirm effectiveness of the proposed method in the cross-platform malware detection with regard to the IoT environment. However, limitations may arise from reliance on the training data availability and need for further evaluation under diverse IoT environments. Another study in [24] proposed an IoT-focused malware classification method using the LCNN (lightweight convolutional neural networks). Initially, malware binaries are converted into the multidimensional Markov images, capturing relationships among the consecutive bytes. LCNN architecture incorporates the depth wise convolution and channel shuffle operations, most importantly minimizing the trainable parameters while maintaining accuracy. Evaluation on the multiple IoT datasets and Microsoft dataset presents the superior performance on the low level features methodologies, accomplishing the average accuracies exceed to 95% on various IoT datasets and 99.356% on Microsoft dataset. LCNN's tiny high accuracy, model size and compatibility across the platforms make it particularly suitable for an IoT environments. Despite the simplicity of the model, it outperforms complex models like VGG16, suggesting its efficacy in IoT security. However, these limitations may include scalability and the adaptability to evolve the malware variants.

Recent work highlights the FL (federated learning) and the deep models as strong defenses against the IoT malware with privacy protection. In [25] a study presented the SIM-FED, a 1D-CNN based FL model trained over IoT-23. It gained the 99.52% accuracy and presented the strong defense against the adversarial attacks. Similarly, another study in [26] proposed federated DNN with regard to the Android-based IoMT. The model accomplished 98.84% accuracy on the real datasets while protecting the user data. Beyond CNNs, the researchers have test GNNs (graph neural networks) in FL settings. In [27] authors presented Fed-MalGAT, a GAT-based model for IoHT. It accomplished 93% accuracy and perform better than Fed-MalGCN, showing strength of attention layers in distributed training using generative models. Lightweight federated GAN model on Aposemat IoT-23 gained 98% accuracy while minimizes the computation requirements [28]. Together, CNN, GNN, and GAN models under FL depicts the clear gains on the centralized training with privacy kept intact. But the issues remain in scaling to the large networks, lowering the communication costs, and supporting different IoT devices. Such gaps highlight the need of continued study.

In [29] a study presents the innovative vision-based AMD (automated Android malware detection) model is presented, employing 16 fine-tuned CNN algorithms. Unlike conventional AMD methods, the model avoids manual feature extraction through converting the bytecodes of 'classes.dex' files into the visual color and grayscale images, thus capturing additional texture details. Using the imbalanced benchmark dataset, this model accomplished 99.40% accuracy with regard to balanced samples and 9805% for imbalanced samples. This model outperforms existing methods, presenting high detection efficiency with low computational overhead. Furthermore, the performance evaluation on both imbalanced and balanced datasets highlights the superiority of the proposed model, especially when utilizing the fine-tuned Xception CNN algorithm. The computational complexity analysis reveals the reasonable storage requirements and the execution times, while comparative analysis highlights the model's superior performance over the recent AMD models. Table 1 shows a summary of some existing techniques.

In a previous study [5], computer scientists have also presented a DL system for detecting malware while integrating dynamic characteristics. These attributes included measures, like packets transmitted/received, the quantity of processes bytes transmitted/received, CPU usage, and memory consumption. The study in [35] used a combination of Deep and Convolutional Neural networks to classify malware, and the results showed that DL models outperformed typical ML models. Notably, regarding malware classification, the CNN methodology outperformed the DNN method. Additionally, Zhang

**Table 1. Summary of existing techniques.**

| Ref. | Feature Types | Approach | Dataset | Performance |
|---|---|---|---|---|
| [14] | Hybrid Static and dynamic | Deep Learning (CNN) model with ML classifiers (SVM) and stacked classifiers | PE malware and benign file samples | 97% |
| [15] | Deep static features | Deep Learning approaches such as LR, ANN, CNN, LSTM, including Transfer Learning (InceptionV3) | BIG15 dataset, images files | 98.76% test accuracy |
| [16] | Static features | Malware visualization method (MCTVD), CNN | Microsoft's malware images | 99.44% |
| [17] | Static features | Enhance AlexNet CNN | Microsoft malware dataset and the Google Code Jam (GCJ) | Microsoft malware dataset 99.99%, and the GCJ dataset reaches 99.38% |
| [18] | Hybrid Static and dynamic | Space-filling curves | PE from the VirusTotal and make dataset and SFC dataset | 97.6% accuracy |
| [19] | Grayscale images 2D | RNNs and CNNs | iNetSim | AUC = 0:96 |
| [20] | Deep features | GRU-SVM, CNN-SVM, MLP-SVM | Malimg Include 9,339 malware images across 25 families | 84.92% accuracy |
| [21] | Image based features are used that are obtained by converting binary files into gray-scale images | Lightweight two-layer CNN | Dataset, obtained from IoTPOT honeypot, includes Classes Linux. Gafgyt, Mirai. | 94.0% accuracy |
| [30] | Local features are used along with global features | Inception3 and RNN | VisDroid include five types of classes | 96% accuracy |
| [22] | Texture features obtained from GIST descriptors is used along with deep features | CNN-SVM | Malimg Include comprises 9,339 samples from 25 malware families | 97.5% accuracy |
| [23] | Assembly, binary and the visible string | Enhanced CNN | Dataset1 contain IoT malware, Dataset2 contain both IoT and traditional malware platform. | 95.31% Acc on Dataset1 98.57% Acc on Dataset2 |
| [24] | Multidimensional based Markov images | LCNN | IoT datasets, Microsoft dataset | 95% Acc on IoT datasets 99.356% Acc on Microsoft dataset. |
| [29] | Automated vision-based android malware detection | AMD model employing with 16 layer CNN | Leopard Android dataset | 99.40% Acc on balanced samples 98.05% Acc on imbalanced samples |
| [31] | Capture the intrinsic properties of malwares with the hierarchical graph | MalGraph, represents executable with hierarchical graphs and uses an end-to-end learning framework for malware detection. | Mixed wild dataset by VirusTotal | 99.97% accuracy |
| [23] | PE malware binaries | LSTM paired with LightGBM in Portable Executable (PE) Malware Classification | SoReL-20M (SOPHOS) | 91.73% accuracy |
| [32] | Hybrid model that combines the traditional PE-32 file features for OS-level actions (i.e., system calls), social networking features and community features | 389 social networking patterns and community detection features | Large dataset with 20 classes | 90.9% accuracy |

*(Continued)*

**Table 1.** (Continued)

| Ref. | Feature Types | Approach | Dataset | Performance |
|---|---|---|---|---|
| [33] | PE file images, adversarial samples, deep features | Hybrid DL + ML, Adversarial Training, One-Class SVM | VirusShare + VXHeaven | 99.06% test accuracy using RegNetY320 + SVM Lowest evasion rate: 12% on VirusShare, 18% on VXHeaven |
| [27] | Graph features from IoHT network (GAT/ GCN-based), federated device logs | Fed-MalGAT (FL + GAT), Fed-MalGCN (FL + GCN) | IoHT malware dataset | Fed-MalGAT: 93% Acc, 92% prec, 93% F1 Fed-MalGCN: 92% Acc, 87% prec, 91% F1 ROC-AUC: 0.926 (Fed-MalGAT), 0.912 (Fed-MalGCN) |
| [26] | Behavioral and network features from Android-based IoMT devices | Federated deep learning + distributed DNN | Two real-world Android malware datasets | 98.84% accuracy in malware detection |
| [34] | Traffic features from IoT devices (malicious code patterns) | GAN-based deep learning over federated architecture | Aposemat IoT-23 dataset | ~98% training accuracy, 97.5% validation accuracy |
| [25] | Network traffic features from IoT devices (lightweight 1D-CNN) | SIM-FED: Federated Learning + deep learning (1D-CNN with FedAvg aggregation) | IoT-23 dataset | 99.52% accuracy; robust against white-box and black-box attacks with minimal degradation |

et al. [36] used a feature engineering method to extract helpful data from API call patterns. For malware classification, they used two models, Gated CNNs and a BiLSTM, with dataset characteristics generated using hashing algorithms on API-named classes and arguments. The assessment of this model, which uses API call patterns as data characteristics, was compared to key findings.

Most of the existing federated malware detection techniques are described in this section which rely on traditional federated averaging (FedAvg) and simple ensemble fusion that cannot handle data distribution across different nodes and classes. In contrast to this, a framework is proposed in this article that aims to overcome the limitations found in previous studies by incorporating the following important improvements. First, it employs a representative malware dataset of 28 classes validated by multiple antivirus engines that reflects the latest malware variations and zero-day attacks. Second, instead of relying on global pre trained model, it implements a hybrid approach for feature extraction that shows the strength of using both static and dynamic analysis techniques, while simultaneously highlighting the shortcomings of earlier methods. Then this hybrid approach is further enhanced by accuracy aware weighting method which prioritizes the high-performance nodes in order to capture changing the malware behavior. The critical evaluation of proposed model and its improvement as compare to the existing method is defined as shown in Table 2.

## 4. Materials and methods

This section highlights the potential methodology for classification of malware families using the federated ensemble and the stacking techniques. The Aim of this proposed technique is to convert the PE documents into binary format, that transform it into the greyscale images for further analysis. The subsequent transformation integrates advantages of both raw binary malware data and the image based DL methods. Once this conversion is complete, method for extracting the detailed feature patterns and representation from binary data is used. The extracted features are used as basis for both the training and evaluating ML model, aiming to recognizing the valuable patterns embedded in PE files. This multistage technique efficiently integrates file format conversion, image generation, and powerful deep based feature extraction that

**Table 2. Comparison of proposed model with existing techniques.**

| Ref(s) | Existing Limitation | Reason/ Drawback | Improvement in Proposed Approach |
|---|---|---|---|
| [14–19] | Centralized dependency | No privacy, low scalability | Federated, privacy-preserving training |
| [16–17] | Static-only features | Miss dynamic behavior | Multi-modal feature fusion |
| [13–24] | No zero-day evaluation | Seen-only performance | Zero-day adaptive ensemble |
| [30–31] | Non-IID sensitivity | Model drift, instability | Class-specialist local training |
| [32–33] | High communication cost | Large model exchange | Quantized, partial updates |
| [20–25] | Dataset-specific tuning | Poor cross-domain transfer | Cross-dataset generalization |
| [26–28] | Simple aggregation (FedAvg) | Ignores client diversity | Adaptive weighted aggregation |

create a robust framework for analyzing and classifying PE files according to their inherent properties. The proposed framework is illustrated in Fig 1.

### 4.1. Dataset creation and compliance

The dataset comprises of 19,600 Win32-type malware samples files accurately collected from various publicly available repositories (e.g., GitHub, VirusShare, MalwareBazaar, Malshare, VX-underground). These samples are was collected using standard browser and downloaded as original executable files that are stored in a single repository under sand box environment. This repository is provided as dataset metadata. Additionally, rigorous filtration is applied on this data by

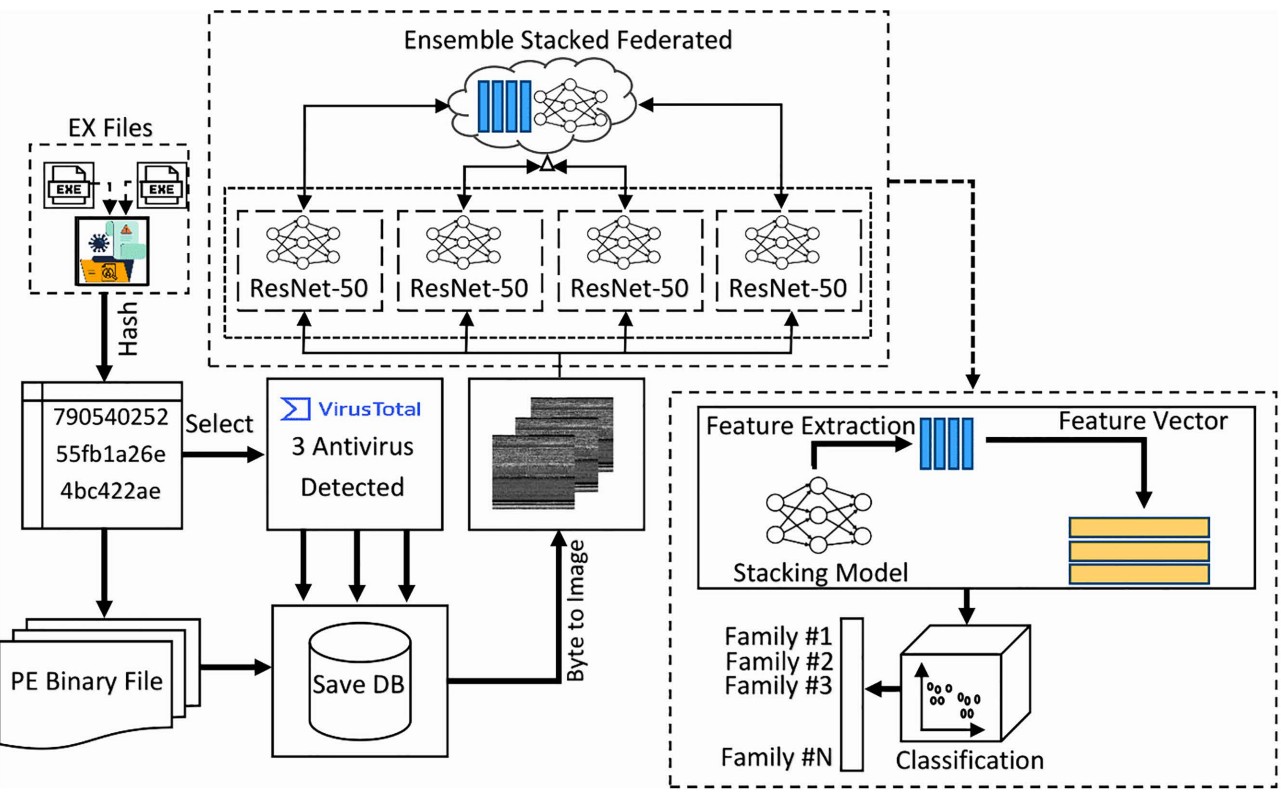

**Fig 1. Proposed model of Malware visualization and classification.**

computing MD5 and SHA256 hashes using standard python library (hash lib). These files are further verified by VirusTotal that remove files with identical hashes. From this metadata analysis file types (ensure Win32-type executables) and antivirus detection names (signatures) were extracted. To maintain labeling consistency, only files flagged by at least three antiviruses are retained. These files were further verified with classes derived from F-Secure virus detection titles, resulting in 28 different classes for classification. MongoDB is used to store all the malware sample files and metadata that prevent data duplication across dataset. These data creation steps are shown in Fig 2.

Malware samples were collected and analyzed under the terms and conditions of all approved data source repositories. All the samples analyze in secure, isolated laboratory systems that ensure no malicious files were executed on product systems or shared during outside. The study followed institutional cybersecurity research rules and complied with national malware handling laws.

## 4.2. Image transformation

The process of transforming malware raw data in bytes into a pixel of a 225×225 image representation consists of several steps, including binary data extraction, transformation, and structure. Starting from malware data in binary format, a section of raw byte data is extracted and labeled as $img_{raw}$. Originally these raw bytes in hexadecimal format, experience the conversion into binary format. Such transformation is represented as:

$$img_{binary} = bin(img_{raw}) \tag{1}$$

In this case, $bin()$ denotes binary conversion. Binary representation, recognized as $img_{binary}$, is now utilized to create image grid with 2 pixel dimensions measuring 225×225. This step can be represented as follows:

$$M_{image} = create\_empty\_image(225, 225) \tag{2}$$

where, $M_{image}$ shows grid like structured image matrix which serves as the initial blank canvas with regard to filling pixel values derived from byte sequence. Final step is to replace pixels with the binary data collected from infection. Pixels are adjusted due to binary digit in $M_{image}$. Operation may be mathematically represented as follows:

$$populate\_image\_pixels(M_{image}, img_{binary}) \tag{3}$$

where, $M_{image}$ shows the details of malware's binary format, with each pixel representing the binary digit. Pixelated image gives a compressed yet interpretable visual representation of virus data. Ordered sequence of the pixels naturally retains

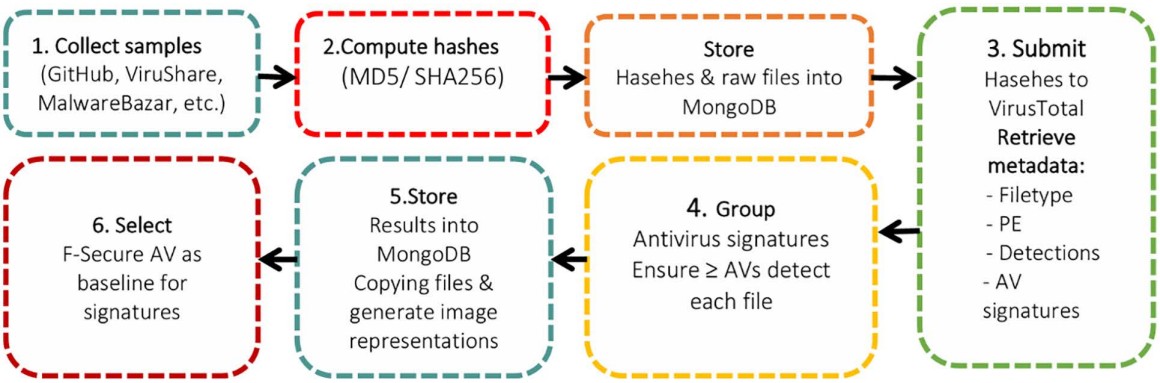

**Fig 2. Dataset creating steps.**

spatial properties that differentiate the binary data, permitting for the analysis of future using image based approaches. Generally, the mathematical representation shows the key steps that are involved in transforming raw binary input data into the structured visual representation appropriate for future analysis and detection in the cybersecurity. This is shown in Fig 3.

In cybersecurity field, recognition of zero day attacks is a fundamental challenge, specially while working with large dataset which consists a collection of malicious attributes. Dataset has 28 distinct classes, each of which shows different features of possible risks regarding security. Respective features are acquired by using architecture of ResNet-18. Every node is responsible with regard to 28 distinct classes, that are mathematically represented as.

$$X_{node} = \{ N_1, \ N_2, N_3 \ldots \ldots \ldots N_n \}$$

(4)

where $X_{node}$ represents set of nodes which contain each element as $N_1, \ N_2, N_3 \ldots \ldots \ldots N_n$ where index $n$ contain total number of nodes.

The training of each class separately is well challenging because of the complexity of task and dependency. The continuous changing nature of zero day attacks makes the traditional methods with regard to the class specific model training poor. In result, a novel method is required to address this difficulty. In order to mitigate this issue more efficiently, collaborative and centralized learning method must be adopted which proves important instead of standalone class wise training.

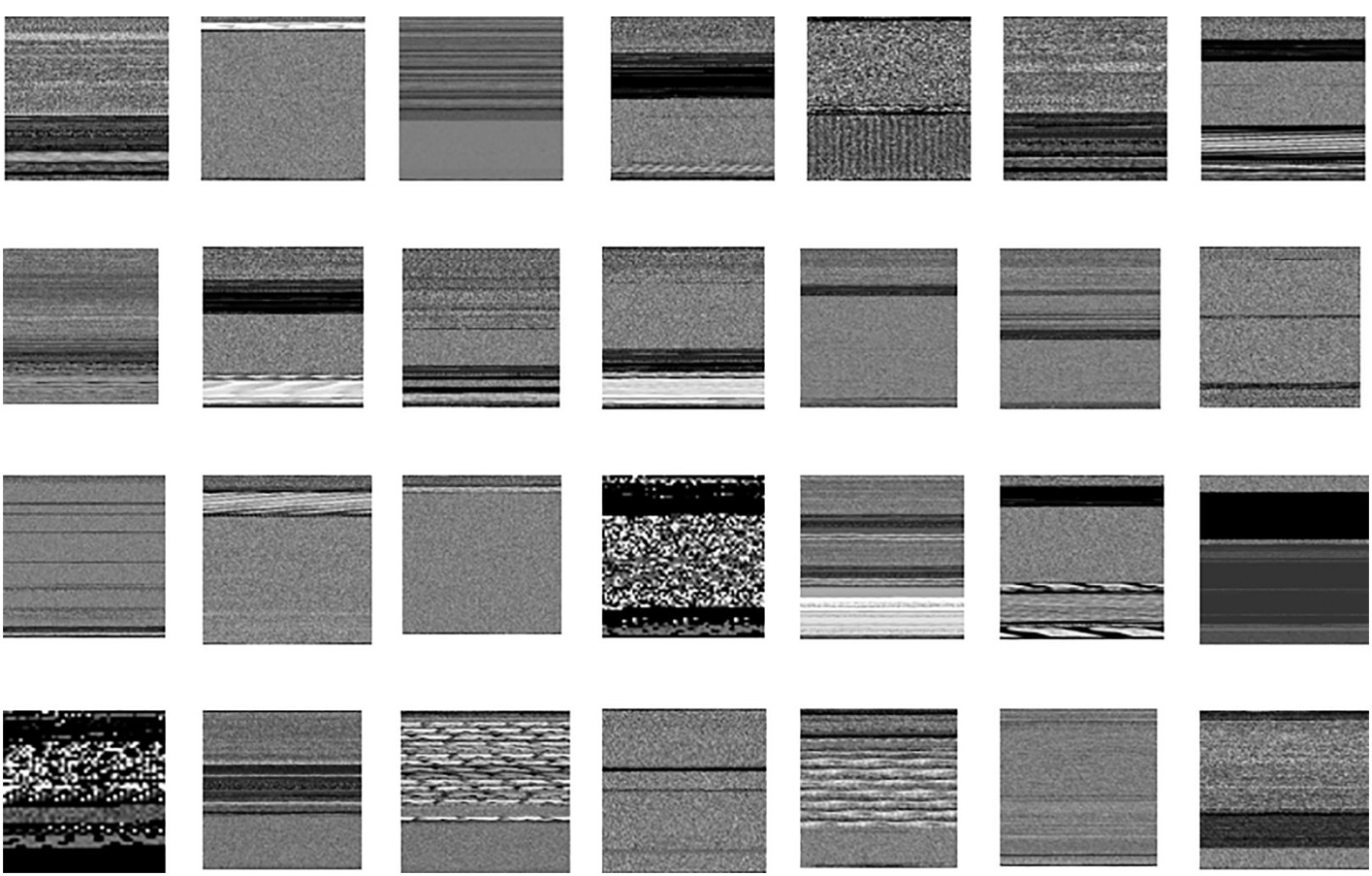

**Fig 3. 28 classes sample images of private dataset.**

FL appears as a promising method, offers real-time, secure, and the distributed model with regard to the training across the multiple nodes. In our proposed system, all 7 nodes are assigned to the sub-class and support the global modeling by utilizing the deep-features extracted by ResNet-18. This decentralized approach permits real time training and secrecy, making it a reliable solution for upcoming modern cybersecurity issues. Objective of this approach is to integrate different nodes into unified framework which responds proactively against new threats in real time. Through employing FL in [37], system maximizes the efficiency of zero day attack detection while make sure the security and privacy of sensitive data. Conversion to FL permits the cybersecurity system to dynamically respond to changing threats, thereby enhancing the system security against previously unknown vulnerabilities. Furthermore, employing collective distributed nodes enhances the ability of system to address new challenges promptly and increases its overall resistance against the cyberattacks. Fundamentally, this method not only enhances the detection abilities while it concurrently strengthens system's defenses in rapidly evolving the cybersecurity domain.

### 4.3. Node-level feature extraction and training

The feature extraction are essential steps with regard to the robust classification. In order to address this, ResNet-18 is implemented to extract the features at the node level across all the nodes. To further compress the feature map for optimized computational efficiency PCA (Principal Component Analysis) is employed that eliminate the redundant features and reduce 512 dimensional feature vector to 256, while maintaining variance 90–95%. PCA gives the advantage in malware image data because of its unsupervised method which enable more accurate classification and detection. In addition, it does not need the additional training and the label information to preserve the variance and to transform the correlated features into orthogonal (uncorrelated) elements. It enhances ability of network to learn the hierarchical and biased features effectively.

System detects the small-scale patterns and the structural attributes related to with the malicious behavior in malware samples. Malware samples are processed by pre-trained ResNet-18 model which learns automatically structural characteristics from input data. Initially, collects low-level information like byte sequences or opcode sequences. Subsequent layers extract more detailed features regarding code's performance. Global average pooling layer condenses such information into a feature vector which captures overall image of malware sample. This pre-trained ResNet-18 model enhances the model's ability to generalize and recognize common malicious features. The extracted deep features are fed into a classifier to detect malware, which makes this model more robust for recognizing and classifying potentially unsafe software. Malware samples are detected by ResNet-18, where $I_{image}$ is the input image H represents height, W represents width of an image, and C represents the number of channels.

$$Con = f(W, I_{image}) + b \qquad (5)$$

where *Con* is the output of the convolutional layer, *f* is the convolutional operation, *W* is the learnable weight and represents the bias value. Representation of the initial convolutional layer is:

$$Con_{(i,j,k)} = \sum_{l.m.n} I_{image}(i+l, j+m, n).W_{l,m,n,k} + b_k \qquad (6)$$

In this equation, *i* and *j* represent the iteration of spatial dimensions *l, n,* and *m* represent the iteration of the filter dimension where k is the index value for the output channel. After the first convolutional layer next step is to normalize $Con_{(i,j,k)}$ to have unit variance and zero mean. Normalization of single batch where mean value is represented as μ and σ represents the standard deviation, whereas β represents the shift and γ represents scale. Representation of batch normalization is:
Representation of batch normalization is:

$$BN(Con) = \frac{con - \mu}{\sigma}.\gamma + \beta \qquad (7)$$

For setting all negative values to zero ReLU activation function is introduced ReLU $(X) = \max(0, X)$ where X represents the matrix in which row represents a data point, and each column represents the output of a neuron or node. After ReLU next block is a residual block which helps in training of federated network by introducing skip connections, output is:

$$R_i(X) = ReLU\left(BN(f(ReLU\left(BN(X, W_1)\right),\ W_2 + X\right) \tag{8}$$

The output of this block is $C_{last} = R_i(X)$ which is now fed into the global average pooling (GAP) represented as:

$$GAP(C_{last})^i = \frac{1}{H \times w} \sum_{j=1}^{H} \sum_{k=1}^{w} C_{last,i,j,k} \tag{9}$$

In this equation, H represents the height and W represents the width of the feature map The global average pooling output feeds into a fully connected layer with weights ($W_{fc}$) and biases ($b_{fc}$), followed by a softmax activation for binary classification. This forms the fully connected layer classifier with softmax activation as:

$$F_{deep} = softmax(W_{fc}.\ GAP(\ (C_{last}) + b_{fc} \tag{10}$$

## 4.4. Ensemble federated model

A distributed network is designed in which multiple entities, nodes, or devices work together to detect malware while retaining a certain degree of control over their local repository. This allows model to be trained across distributed devices without learning raw data, ensuring privacy. The central server initially computes the average of the received model parameters, producing a global average model as a combination of the local models. This training is repeated over multiple epochs to refine the global model. However, simple parameter averaging (as in FedAvg) is not sufficient because data across node is highly imbalanced as it dominates major malware families while minority families remain under-represent. Therefore, in this case, pure averaging with hierarchical stacking ensemble mechanism that integrated level predictions through meta-learners and weight each node according to its accuracy, this amplifies noise from weaker nodes and ensure reliable node contribution for model decision.

The mathematical framework uses a weighted average of model parameters to create a reliable and robust model. Features $F_{deep}$ extracted from ResNet-18 trained on the local nodes are represented as:

$$F_{deep(i)} = ResNetFeatureExtractor(n_i,\ d_i) \tag{11}$$

Local features are extracted which are transferred to a central server. Using $F_{deep(i)}$ each local device trains a local node as:

$$n_i:\ \theta_i = argmin_\theta \frac{1}{|d_i|} \sum_{(x,y)\epsilon\ d_i} \mathcal{L}(n_i(F_{deep(i)}; \theta), y) \tag{12}$$

In this equation, $n_i$ represent the local node train on $i_{th}$ device whereas $\theta_i$ represent the weight (parameters) trained on the local node $n_i$. The value minimum value of $\theta$ is defined as $argmin_\theta$. The average of the local dataset $d_i$ is defined as $\frac{1}{|d_i|} \sum (x, y)\epsilon\ d_i$ whereas the size of the dataset is $|d_i|$. The loss function $\mathcal{L}(m_i(F_{deep(i)}; \theta), y)$ represents the difference between true label $y$ and the node prediction. After completion of the local node training, the parameters $\theta_i$ are sent to the central server where the weighted average is employed for global node parameter $\theta_{avg}$ training designed as:

$$\theta_{avg} = \frac{1}{n} \sum_{i=1}^{n} w_i.\theta_i \tag{13}$$

where $n$ is the number of nodes and $w_i$ assign to each node. The averaged parameters are used to create a global model:

$$Model_{global} : \theta_{global} \leftarrow \theta_{avg} \tag{14}$$

Based on accuracy, the updating method for the weight can be characterized as follows:

$$w_i^{t+1} = \frac{Accuracy\,(n_i^{(t)})}{\sum_{j=1}^{n} Accuracy\,(n_j^{(t)})} \tag{15}$$

where $w_i^{t+1}$ represent the weight of $i_{th}$ node at iteration $t+1$, whereas $Accuracy\,(n_i^{(t)})$ represent the accuracy of $j_{th}$ node at iteration $t$, accuracy of $i_{th}$ node at iteration $t$ is defined as $Accuracy\,(n_j^{(t)})$. The weights $w_i^{t+1}$ are updated based on the local nodes' relative accuracies. If a local node is more accurate than the other localized nodes, its weight will be updated about the others. This suggests that it will have a greater impact on the entire model's evolution in the subsequent iteration. The weight update strategy prioritizes local nodes that did well in the previous phase. This method increases the overall effectiveness of the federated learning.

In this presented model, we introduced new ensemble prediction method for federated learning instead of average global model for learning and prediction instead of global averaging. The ensemble prediction method can be represented as let $M_{node\_model}$, $I_{test}$ as the test input images and $P_i$ as the predcitions node level trained models for the test set input images ensemble prediction can be describe as:

$$P_i = argmax\,(Model_i(I_{test})),\ \forall_i \in M \tag{16}$$

These the mathematical formulation well defines the federated aggregation and ensemble stacking process. To ensure the reproducibility of proposed model, the training and aggregation steps are described in detail. Each client node trains a ResNet-18 model on its local data, and share model weights and softmax predictions to the server. Unlike standard federated averaging or majority voting, the server concatenates all predictions into a logistic regression meta-learner that serves as stacking layer. This stacked ensemble enables the global model to learn complementary decision rules across all heterogeneous nodes to improve robustness and generalization. Additionally, an algorithm is outlined that defines the workflow in a reproducible manner

| Phases | Description |
|---|---|
| Input | Malware dataset $D$ split across $K$ nodes $\{D_1,\ldots,D_k\}$;<br>Base model: ResNet-18;<br>Meta-learner $S$ Logistic Regression;<br>Communication Rounds $T$;<br>Local epochs $E$. |
| Output | Final stacked ensemble model $S$. |
| **Training** | |
| Step 1 | Initialize local models $M_k$ for each node $k \in \{1, 2, \ldots, k\}$. |
| Step 2 | **For** each communication round $t = 1 \rightarrow T$: |
| Step 3 | **For** each node $k$ in parallel: |
| Step 4 | Train $M_k$ on local dataset $D_k$ for $E$ epochs using:<br>• Batch size = 64<br>• Optimizer = Adam (learning rate = 0.001, weight decay = 1e-5)<br>• Loss = categorical cross-entropy |
| Step 5 | Generate prediction probabilities $D_k$ on local validation set. |
| **Communication** | |
| | Send model parameters $\theta_k$ and predictions $P_k$ to central server. |
| End For (all clients have uploaded) | |

| Phases | Description |
|---|---|
| **Stacking** | |
| Step 1 | At the central server: Collect predictions $\{P_1, \ldots, P_k\}$. |
| Step 2 | Build stacked feature matrix F = concat $(P_1, \ldots, P_k)$. |
| Step 3 | Train or update meta-learner $S$( *logistic regression classifier*) on stacked features $F$. |
| Step 4 | Evaluate ensemble performance on validation set. |
| End For (round ***t*** complete) | |
| Return final meta-learner ***S*** as the global stacked ensemble model. | |

There are several advantages of this approach, 1) Each model is trained independently and does not rely on a centralized server for model updates or training data, 2) It also allows models to be trained and updated in parallel, leading to potential improvements in efficiency and scalability, 3) the approach operates independently, there is no need for frequent communication between the central server and clients, 4) if one model fails or experiences issues, it does not affect the operation of other models, 5) there is no need for frequent communication between the central server and clients. Graphical representation of the staking-ensemble model is depicted in Fig 4.

## 5. Experimental setup and results

In this section, the proposed method is compared with several state-of-the-art malware detection and classification methods. The results are evaluated both qualitatively and quantitatively to validate the effectiveness of proposed malware classification approach. Additionally, this system is developed to examine the findings of other conventional methods. These experiments are designed to examine the efficacy of suggested malware detection and classification approach. Furthermore, in this study, the models were evaluated using performance metrics. These methods are analyzed collectively rather than separately. For each iteration, mean precision and accuracy are computed, and then the average F-score is calculated.

### 5.1. Dataset description

The collected malware files are converted into grayscale image files and normalization was applied during preprocessing to convert all images into 255 × 255 dimensions that allows multiple analysis perspectives. Moreover, strict hash-based

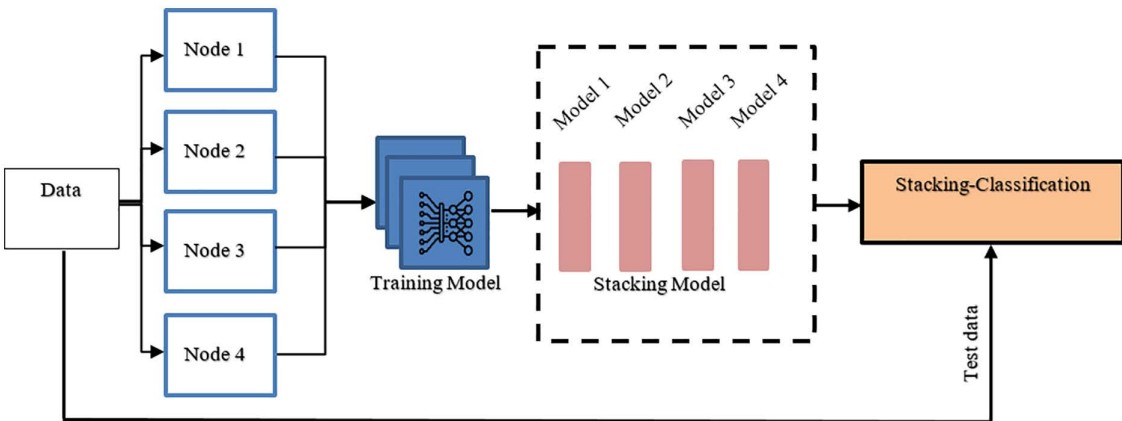

**Fig 4. Graphical representation of the staking-ensemble model.**

**Table 3. Private dataset description.**

| Dataset | Size | Classes | Pre-processing |
|---|---|---|---|
| Training set | 14,000 | 28 | Normalization |
| Validation set | 2,800 | 28 | Normalization |
| Test set | 2,800 | 28 | Normalization |

splitting technique is used for data splitting that prevent data leakage across dataset subsets. The dataset is split into three subsets, 70% for training that contains 14,000 samples within 28 classes, 15% for validation containing 28,00 samples, and similarly for testing as defined in Table 3. This image-based framework supports the development of advanced malware detection technique that prevents malware attacks.

To ensure the robustness of propose model, key training hyper parameters are selected such as 0.001 learning rate and weight decay = 1e-5 with Adam optimizer over 100 epochs and 64 batch size and experiments were also repeated using 5-fold cross validation with categorical cross-entropy as an objective loss function.

This experiment is conducted on an intel Core i9 processor with 64 GB RAM and NVIDIA RTX 3090 GPU (24 GB) in laboratory setting. The model implementation and federated setup is stimulated in Python 3.10 with PyTorch 2.0 with multiple client nodes, each containing a subset of malware families on single workstation with central server. Communication between nodes and server over a local network with synchronous updates in each round. Each round utilized five local epochs per node before the server performed aggregation. Overall, 20 communication rounds were required where each node exchanged about 24 MB parameters per update and total cost across 20 rounds was approximately 1.9 GB for just 4 nodes. In this communication, raw data is not shared because it loses some gradients and parameters information. To overcome this problem, differential privacy (DP) aggregation (SA) protocols are integrated such as Google's RAPPOR, Apple's and Secure Federated Aggregation (SFA) such as Google FL that ensures results reproducibility and scalability for malware detection tasks.

### 5.2. Cross-dataset generalization

This model is trained on a private dataset, which contains 28 different classes, and evaluated on 25 different classes from the Malimg dataset. The performance of different DL models and the proposed Ensemble Federated model is illustrated in Table 4. This table demonstrates that EfficientNet-B0, DenseNet-121, and ViT-B/16 achieve competitive accuracy as compared to ResNet-50 but require higher FLOPs and inference time. Whereas, ResNet-50 reaches 93.1% accuracy with lower 4.1 G FLOPs and fater inference about 7.9 ms per image. In contract, EfficientNet-B0, DenseNet-121, ViT-B/16, and FedAvg take about (8.1 ms/img), 9.5 ms/img, 14.8 ms/img, and about 8.5 ms/img. This highlights the reason why ResNet-50 is selected as the backbone for proposed model federated ensemble framework.

**Table 4. Cross-dataset generalization.**

| Model | Accuracy (%) | F1-Score | AUC | Params (M) | FLOPs (G) | Inference (ms/img) |
|---|---|---|---|---|---|---|
| AlexNet | 89.7 ± 0.6 | 0.86 ± 0.01 | 0.92 ± 0.01 | 61.0 | 0.72 | ~3.5 |
| ResNet-18 | 91.6 ± 0.4 | 0.89 ± 0.01 | 0.94 ± 0.01 | 11.7 | 1.8 | ~5.2 |
| ResNet-50 | 93.1 ± 0.3 | 0.91 ± 0.01 | 0.96 ± 0.01 | 25.6 | 4.1 | ~7.9 |
| EfficientNet-B0 | 93.8 ± 0.5 | 0.92 ± 0.01 | 0.96 ± 0.01 | 5.3 | 0.39 | ~8.1 |
| DenseNet-121 | 94.1 ± 0.4 | 0.92 ± 0.01 | 0.96 ± 0.01 | 8.0 | 2.8 | ~9.5 |
| Vision Transformer (ViT-B/16) | 94.5 ± 0.4 | 0.92 ± 0.01 | 0.96 ± 0.01 | 86.0 | 17.6 | ~14.8 |
| FedAvg (ResNet-18) | 92.2 ± 0.4 | 0.90 ± 0.01 | 0.95 ± 0.01 | 11.7 | 1.8 | ~5.5 |
| Proposed Model | 95.4 ± 0.2 | 0.93 ± 0.01 | 0.97 ± 0.01 | $25.6 \times K + 0.1$ (meta) | ~4.1/node | ~8.5/node |

Additionally, an ablation study is presented in Table 5 that highlight the improvement by employing stacked federated with model ResNet-50 backbone. Although EfficientNet-B0, DenseNet-121, Vision Transformer (ViT-B/16), and FedAvg baseline provide competitive results but these models demand high computational cost. In the FL, communication requires lower FLOPs and faster inference time. Therefore, ResNet-50 is adopted as primary backbone for all subsequent results.

### 5.3. Experiment 1: Class-wise performance evaluation

This experiment is conducted to evaluate the model's performance on each malware family, as shown in Table 6, in terms of accuracy, F1-score, precision, and recall. Accuracy defines the overall model performance in correctly identifying malware; it mainly focuses on true positives and true negatives. Whereas, precision is used to check how many of

**Table 5. Ablation study model backbone and Fusion on private and Malimg dataset.**

| Setting | Backbone | Fusion | Acc (%) | F1 | AUC |
|---|---|---|---|---|---|
| Single model | ResNet-50 | – | 93.1 | 0.91 | 0.96 |
| FedAvg | ResNet-50 | Param avg | 92.2 | 0.90 | 0.95 |
| Proposed | ResNet-50 | Stacking (logistic meta-learner) | 95.4 | 0.93 | 0.97 |

**Table 6. Class-wise performance on Private dataset.**

| Malware Family | Malware Name | Precision | Recall | F1 Score | Support |
|---|---|---|---|---|---|
| Adware | Adware_ADWARE_Imali_Gen | 0.97 | 0.98 | 0.96 | 120 |
| | Adware_ADWARE_InstallRex_Gen | 0.98 | 0.97 | 0.97 | 115 |
| | Adware_ADWARE_InstMonster_Gen7 | 0.97 | 0.97 | 0.96 | 110 |
| | Adware_ADWARE_MultiPlug_Gen7 | 0.98 | 0.98 | 0.97 | 125 |
| | Gen_Variant_Adware_MultiPlug | 0.97 | 0.98 | 0.96 | 105 |
| | PotentialRisk_PUA_Multiplug_aoa | 0.96 | 0.97 | 0.96 | 98 |
| | Adware_Mplug_LO | 0.98 | 0.97 | 0.97 | 120 |
| | Gen_Variant_Adware_Mplug | 0.97 | 0.98 | 0.96 | 117 |
| | Gen_Adware_BrowseFox_1 | 0.97 | 0.98 | 0.96 | 108 |
| | Gen_Variant_Adware_Graftor | 0.97 | 0.97 | 0.96 | 113 |
| | Gen_Variant_Adware_Kazy | 0.97 | 0.98 | 0.96 | 100 |
| | Gen_Variant_Application_Bundler | 0.96 | 0.97 | 0.96 | 103 |
| | Gen_Variant_Application_Kazy | 0.97 | 0.98 | 0.96 | 111 |
| | PotentialRisk_PUA_Outbrowse_Gen | 0.97 | 0.97 | 0.96 | 109 |
| | PotentialRisk_PUA_Somoto_Gen2 | 0.97 | 0.98 | 0.96 | 106 |
| Trojan | Trojan_Agent_BDMJ | 0.89 | 0.9 | 0.88 | 95 |
| | Trojan_TR_Agent_nmczf | 0.91 | 0.89 | 0.9 | 90 |
| | Trojan_TR_Crypt_xPACK_Gen | 0.9 | 0.91 | 0.89 | 87 |
| | Trojan_TR_Dropper_Gen | 0.92 | 0.9 | 0.91 | 93 |
| | Trojan_W32_Cryptolocker_C | 0.9 | 0.89 | 0.89 | 85 |
| Worm | Email_Worm_W32_Mydoom_gen_A | 0.84 | 0.85 | 0.83 | 77 |
| | Net_Worm_W32_Allaple_gen_B | 0.83 | 0.84 | 0.82 | 70 |
| Win32 Malware | Win32_Parite_B | 0.8 | 0.81 | 0.79 | 68 |
| | Win32_Parite_C | 0.81 | 0.8 | 0.8 | 66 |
| | Win32_Ramnit | 0.79 | 0.8 | 0.78 | 72 |
| | Win32_Sality_3 | 0.8 | 0.79 | 0.79 | 64 |
| | Win32_Virlock_Gen_1 | 0.81 | 0.79 | 0.8 | 63 |
| | Win32_Virtob_Gen_12 | 0.79 | 0.81 | 0.78 | 69 |

the detected malware samples are correct. On the other hand, recall measures how many actual malware samples are detected by the system. The F1-score assesses the balance between recall and precision.

Experimental results show that the model performs well on some malware classes, such as Adware family achieved a higher score, above 0.96. Whereas, there is some variation in the performance of the Win32 malware family, showing overall results around 0.79 to 0.81. These changes can be because of factors like intra-class variations, class imbalance, and the feature overlapping. It shows that this malware family is much more challenging to detect.

Fig 5 presents confusion matrices for Adware, Trojan, Worm, and Win32 malware families. Each matrix shows true positives (TP), false positives (FP), false negatives (FN), and true negatives (TN) for each respective family. Proposed model achieves highest precision and recall above 90% for Adware and Trojan families, these highlight accurate detection of model. Worm and Win32 families show comparatively lower results around 80–85%, that make these families more challenging to classify. The separate matrices show that detection strength differs across groups and highlight the need to improve handling of minority or overlapping families.

Fig 6 presents ROC curve plotted using True Positive Rate (TPR) against False Positive Rate (FPR) for Adware, Trojan, Worm, and Win32 malware families. Each curve provides a specific threshold for diagnosing model performance in detecting a particular malware family. Adware achieved the strongest performance with an AUC around 0.97, that separate the positive and negative rates. Trojan detection also performed strongly, with an AUC of 0.94 and very low false positives. Worm detection reached an AUC of 0.92, maintaining strong accuracy but with slightly reduced recall compared

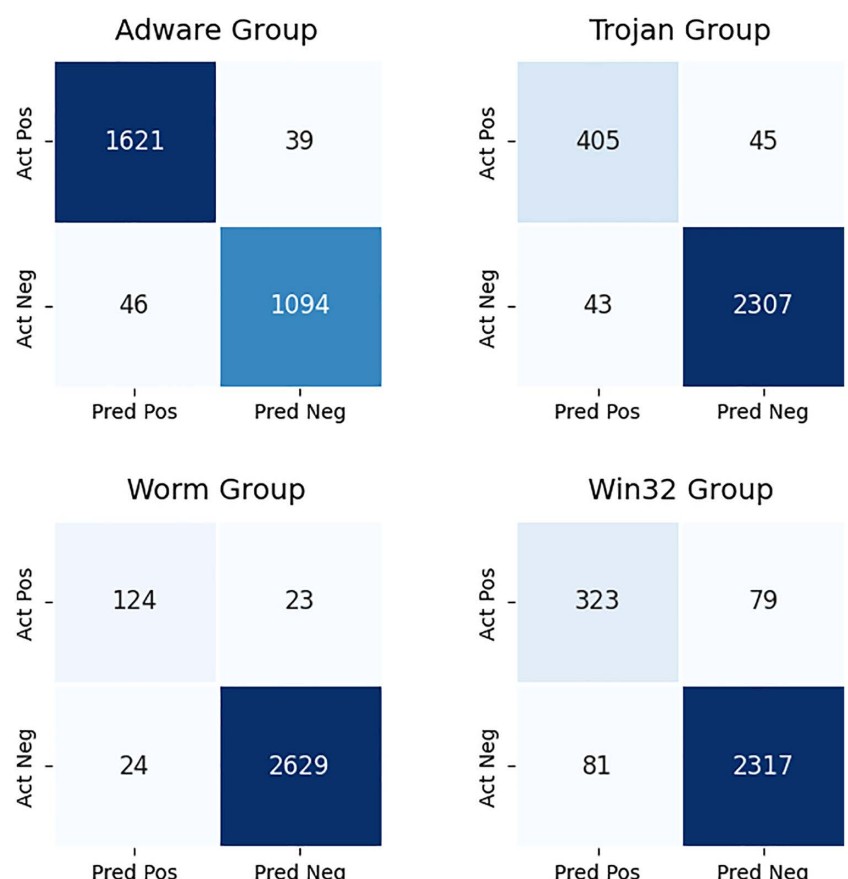

**Fig 5. Family-wise confusion matrices for Adware, Trojan, Worm, and Win32 families.**

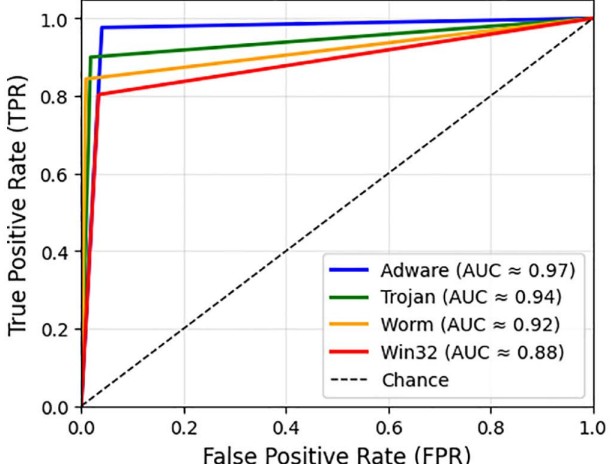

**Fig 6. ROC curve comparison across Malware Families.**

to Adware and Trojan. Win32 scored the lowest, with an AUC of 0.88, showing weaker sensitivity and more false positives. On the whole, the ROC analysis depicts reliable classification across the entire group. The Adware and Trojan had the most stable detection; on the other hand, Win32 remained more challenging because of polymorphic and obfuscated samples.

### 5.4. Experiment 2: Meta-federated learning training and validation

It is important to take note of the Meta-Learner Adaptation Metrics for an overall assessment of the performance of the proposed framework of federated learning. The analyses in this chapter on the complex dynamics of the federated learning process support detailed insights into the meta-learner adaptation at the individual nodes. Ensemble federated learning used in this work offers a strong and privacy-preserving meta-learning approach over a distributed network. The learning curves of the proposed meta-federated learning model for malware detection are depicted in Fig 7 by representing the performance of different epochs at key metrics. The "Validation Loss" shows the training loss of the model, indicating how well the model fits the training data. On the other hand, the "Accuracy" represents the training accuracy, which is defined as the percentage of examples properly classified during training. Besides that, "Validation" provides insight into the model's capability to generalize to data that has not been seen previously.

This is the most crucial part of the evaluation to establish the framework. It can successfully apply its obtained information to new cases. Hence, improving robust performance. Tracking these parameters over epochs enables the continuous assessment of the model's training history. During early epochs, training loss often decreases with the rise in training accuracy. This shows the model's capacity to adapt to its training data. Meanwhile, the validation measures indicate how effectively the model applies to previously unknown data to prevent overfitting.

Both graphs in Fig 7 show the change of a value over a set number of epochs. These epochs are repetitive throughout the data training. In the Meta-Learning Loss Across Epochs graph, the y-axis represents loss, which indicates the task performance model. Where lower values represent higher performance. The x-axis indicates epochs, and the line graph shows a decrease from roughly 0.9 to about 0.75 as epochs proceed, demonstrating that the model performs better as more data is given. In Meta-Learning Accuracy graph, the y-axis displays accuracy, which is the model's

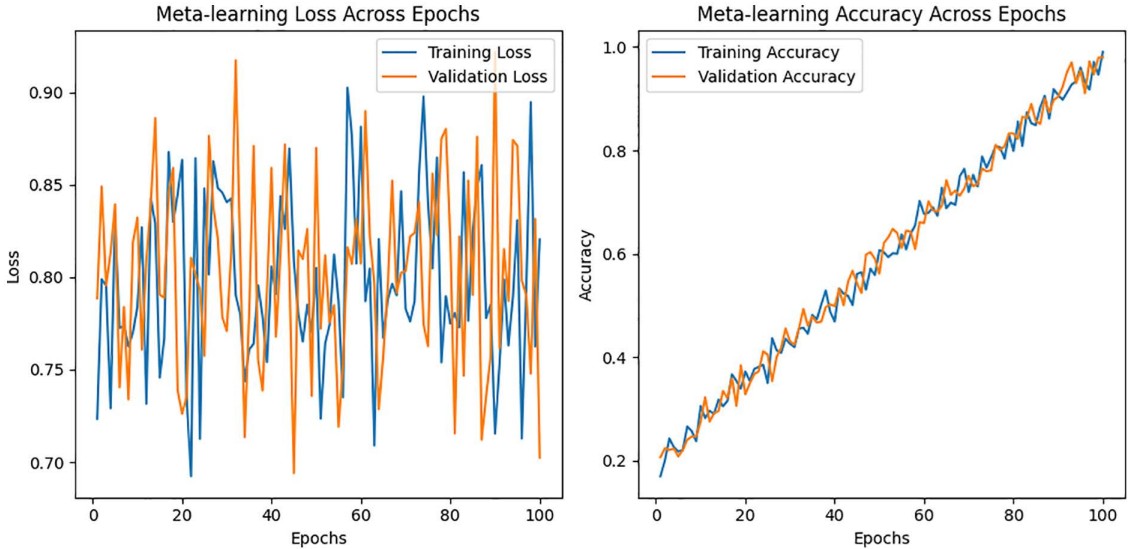

**Fig 7. Meta learning performance across Epochs.**

proportion of true predictions. High accuracy numbers suggest higher performance. Epochs again represent the x-axis, and the line graph shows a rise from around 0.7 to almost 1 as epochs go, indicating increased accuracy as the algorithm learns from more data.

### 5.5. Experiment 3: Baseline model comparison using quadratic SVM classifier

In this experiment, baseline models are applied to all 28 classes to validate. In Table 7 explain the results of Class wise accuracy of models on the Quadratic SVM classifier. The table examines the performance of deep learning-based classification models. ResNet-50 is identified as the most accurate model. While AlexNet stands out as the most exact. This difference frequently involves a trade-off that changes depending on the particular application at hand. For instance, emphasizing false negatives may lead to ResNet-50, whilst assuring the importance of positive classifications may lead to AlexNet.

To evaluate the model's efficiency, three key performance measures are used, such as False Negative Rate, often known as FNR. It is the percentage of negative incidents that were incorrectly categorized. Accuracy refers to the model's capacity to produce accurate predictions. In the analysis, AlexNet has the lowest accuracy of 90.90%, whereas ResNet-50 has the best accuracy of 94.8%. Precision refers to the number of correct predictions. In this comparison, AlexNet achieves a high precision of 90.97%, while ResNet-50 has the lowest precision at 94.29%.

**Table 7. Class Class-wise accuracy of baseline models using Quadratic SVM classifier.**

|  | FNR | Accuracy | Precision |
|---|---|---|---|
| **AlexNet** | 9.10±0.3 | 90.90±0.4 | 90.97±0.3 |
| **EfficientNet-B0** | 8.79±0.2 | 91.41±0.4 | 91.71±0.2 |
| **ResNet-18** | 8.79±0.2 | 91.41±0.4 | 91.71±0.2 |
| **ResNet-50** | 5.20±0.2 | 94.80±0.3 | 94.29±0.2 |

### 5.6. Experiment 4: Zero-day attack performance

This experiment is conducted to evaluate the strength of the model against zero-day attacks, which are new malware threats that cannot be detected easily. To stimulate real conditions, some malware families such as Graftor, Kazy, and BrowseFox (Adware), Zbot and Agent (Trojan), and Mydoom (Worm) were not included in training but used as unseen malware during testing. This ensures the model's efficiency results can detect unseen threats rather than simple recognition of known families.

Table 8 shows the behavior of different malware families and the performance of various deep learning pretrained models against these attacks. The table includes similar performance measures along with the detection rate, which shows the percentage of correctly identified malware samples for each model.

Experimental results show that among all traditional deep learning models, proposed ensemble models perform better and achieve the highest overall performance with 94.6% detection rate, 0.91 F1-Score that highlighting the importance of integrating multiple models to defend against zero-day attacks. This improvement is achieved from the hierarchical stacking method, which utilized node diversity and integrate predictions from multiple nodes. This experiment also supports the model to handle new malware variants and demonstration stronger zero-day detection than standard federated averaging.

Another experiment is performed to validate the model ability to detect zero-day malware attacks. Leave-one-Family out protocol is applied where each malware family was excluding from training and considered as unseen data during testing. This test was repeated for all families and averaged over five random seeds. Table 9 reports the results: the ensemble reached a 94.6±0.4% detection rate and 0.91±0.01 F1-score. These findings show the model maintains strong accuracy when facing previously unseen malware families.

**Table 8. Zero-day attack detection on Malware Families.**

| Malware Family (Zero-Day) | Zero-Day Attack(s) | Model | Detection Rate (%) | F1-Score | Precision | Recall |
|---|---|---|---|---|---|---|
| Adware | Graftor, Kazy, BrowseFox | ResNet-18 | 88.4 | 0.86 | 0.89 | 0.84 |
| | | ResNet-50 | 91.0 | 0.88 | 0.92 | 0.85 |
| | | AlexNet | 84.2 | 0.83 | 0.85 | 0.81 |
| Trojan | Zbot, Agent, Dropper | ResNet-18 | 87.1 | 0.84 | 0.87 | 0.82 |
| | | ResNet-50 | 89.5 | 0.86 | 0.9 | 0.83 |
| | | AlexNet | 82.7 | 0.81 | 0.83 | 0.79 |
| Worm | Mydoom | ResNet-18 | 89.2 | 0.85 | 0.88 | 0.83 |
| | | ResNet-50 | 92.0 | 0.89 | 0.93 | 0.86 |
| | | AlexNet | 86.4 | 0.82 | 0.84 | 0.8 |
| | | Ensemble Fed | 94.6 | 0.91 | 0.94 | 0.89 |
| Win32 Malware | Sality, Virut, Parite | ResNet-18 | 86.7 | 0.83 | 0.86 | 0.81 |
| | | ResNet-50 | 90.1 | 0.87 | 0.91 | 0.84 |
| | | AlexNet | 83.2 | 0.8 | 0.83 | 0.78 |

**Table 9. Zero-day detection with leave-one-family-out protocol (averaged over 5 seeds).**

| Held-Out Family | Training Families | Testing on | Detection Rate (%) | Best Model |
|---|---|---|---|---|
| **Adware** | All Adware except Graftor + all Trojans + Worms + Win32 | Graftor samples | 87.4±0.7 (avg) | ResNet-50 (90.5±0.5) |
| **Trojan** | All Trojans except Zbot + all Adware + Worms + Win32 | Zbot samples | 86.0±0.6 (avg) | ResNet-50 (89.2±0.5) |
| **Worm** | All Worms except Mydoom + all Adware + Trojans + Win32 | Mydoom samples | 88.7±0.5 (avg) | ResNet-50 (91.8±0.4) |
| **Win32** | All Win32 except Sality + all Adware + Trojans + Worms | Sality samples | 86.2±0.7 (avg) | ResNet-50 (89.8±0.6) |
| **Overall** | Leave-one-family-out across all 28 families | All 28 families | 87.1±0.6 (avg) | **Ensemble Fed (94.6±0.4)** |

### 5.7. Experiment 5: Classifiers comparison with private dataset

In this experiment, extracted features are compared using different classifiers. This experiment demonstrates that the best classifier option differs based on the measure used. In this regard, Quadratic SVM outperforms all kernel types in terms of AUC performance, demonstrating its ability to identify positive from negative examples, but the Decision Tree has the least FRN values, showing its ability to reduce missing negative cases across all kernel types. The comparison of multiple classifiers is shown in Table 10 on a private dataset. The comparison mainly focuses on deep learning models over the multiple classifiers like linear, quadratic and the cubic SVM and decision trees. The model performance is evaluated by using FRN and AUC, as well as accuracy. Notably, decision tree has lowest FRN of 17.7%, while the Cubic SVM has highest 27.0% accuracy with regard to the Linear SVM and the lowest 3.1% accuracy for the Cubic SVM. The AUC statistic measures the model's capacity to prioritize positive instances over negative instances. Higher values for the AUC suggest better performance. Quadratic SVM has the greatest AUC of 94.2% for linear SVM and 97.1% for quadratic SVM. Whereas linear SVM has the lowest AUC of 88.0% for linear SVM and cubic SVM achieve 84.00% AUC.

### 5.8. Experiment 6: Node level performance

In this experiment, node performance was evaluated using multiple deep models. The evaluation included computing important performance measures. Which including F-Score, Accuracy of the model, Sensitivity rate, and AUC. By doing node-level analysis, each model's effectiveness in processing the provided datasets is obtained. The study results provide useful information about the relative performance of the various mode ls. This information allows for intelligent decisions about their applicability for various applications or objectives in machine learning contexts. Table 11 describe the node level performance of the proposed model.

The performance of these models is calculated using these measures such as accuracy, precision, and the F1 score. ResNet-50 has the best accuracy among all four nodes (92.86% to 94.57%), whereas AlexNet shows the lowest accuracy (73.43% to 94.57%). AlexNet has the most precision for three of the four nodes (92.00% to 94.57%), while ResNet-50 has the least precision (92.86% to 94.29%) for all three nodes. ResNet-18 and ResNet-50 have identical F1 values across all four nodes. However, AlexNet has lower F1 scores. ResNet-50 consistently obtains the best accuracy, suggesting greater overall classification. In contrast, AlexNet often has

Table 10. Performance of Private dataset on deep model and classifiers.

| Model/ Classifiers | AlexNet | | | ResNet-18 | | | ResNet-50 | | |
|---|---|---|---|---|---|---|---|---|---|
| | FRN | AUC | Accuracy | FRN | AUC | Accuracy | FRN | AUC | Accuracy |
| Linear SVM | 8.9 | 88.0% | 91.1% | 7.0 | 90.0% | 92.0% | 3.6 | 84.00% | 96.40% |
| Quadratic SVM | 5.8 | 92.0% | 94.2% | 3.6 | 84.0% | 96.4% | 2.9 | 87.00% | 97.10% |
| Cubic SVM | 27.0 | 80.0% | 73.0% | 2.9 | 87.0% | 97.1% | 3.1 | 83.60% | 97.40% |
| Decision Tree | 17.7 | 92.1% | 82.3% | 12.3 | 93.1% | 87.7% | 3.5 | 86.00% | 96.60% |

Table 11. Performance metric of AlexNet, ResNet-18 and ResNet-50 on each node using Custom dataset.

| Node/ Model | AlexNet | | | ResNet-18 | | | ResNet-50 | | |
|---|---|---|---|---|---|---|---|---|---|
| | Accuracy | F-score | Precision | Accuracy | F-score | Precision | Accuracy | F-score | Precision |
| Node 1 | 94.57% | 94.47% | 94.57% | 92.00% | 91.96% | 92.00% | 94.57% | 94.47% | 94.57% |
| Node 2 | 80.71% | 80.41% | 80.71% | 87.57% | 87.53% | 87.57% | 73.43% | 73.36% | 73.43% |
| Node 3 | 95.29% | 95.28% | 95.29% | 98.71% | 98.71% | 98.71% | 92.86% | 93.01% | 92.86% |
| Node 4 | 79.14% | 74.46% | 79.14% | 91.00% | 90.96% | 91.00% | 65.29% | 60.06% | 65.29% |

the best accuracy, meaning a larger possibility of true positive classifications. ResNet-18 and ResNet-50 have comparable F1 scores. Proposing a balanced outcome in terms of accuracy and recall. Model selection should be based on the unique application needs. ResNet-50 may be recommended for reducing false negatives because of its excellent accuracy. While AlexNet may be a better option for establishing really meaningful positive classifications due to its improved accuracy. Fig 8 represents the node level result of confusion matrix on two deep learning model ResNet-50 and AlexNet Multiple deep-based models are used for comparison examination. ResNet-18 appeared as the best performer, with exceptional performance indicators. With a remarkable 98% accuracy ResNet-18 beat other models. Due to its higher performance, ResNet-18 was chosen as the recommended model across all 28 nodes. This decision emphasizes the model's resilience and efficacy in tasks or dataset handling under consideration. Choosing ResNet-18 indicates its dependability and suitability for implementation in a variety of settings. This highlights its capacity to give optimal outcomes across several applications or domains.

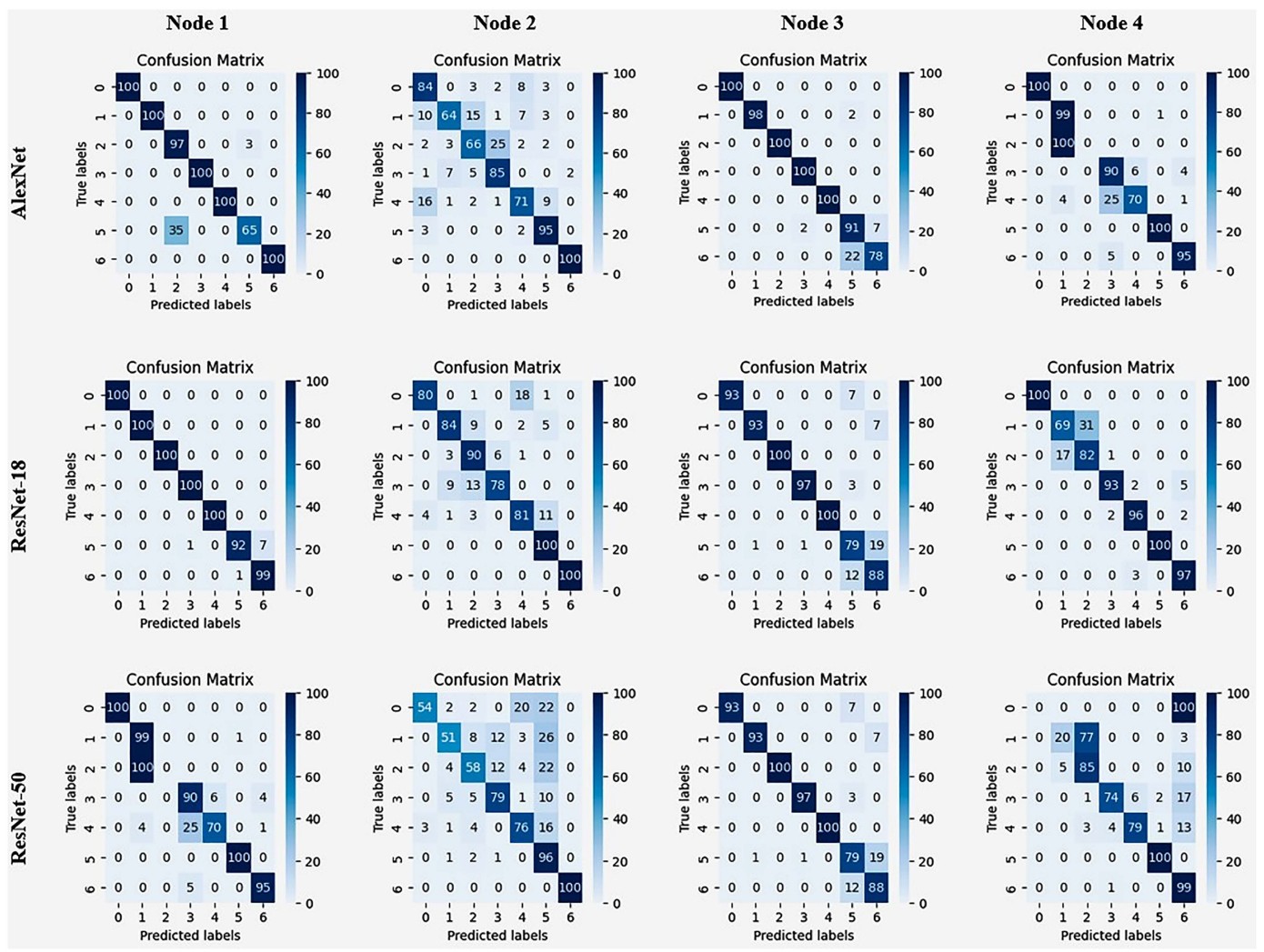

**Fig 8. Four different confusion matrix results on AlexNet and ResNet-18.**

## 5.9. Node level performance on malimg dataset

The Malimg dataset [38] is a publicly accessible dataset that is often used for malware classification. It contains 9339 images that depict various varieties of malware. The images are divided into 25 unique classes, with each class representing a certain form of malware. This dataset is mainly utilized to train and test ML models for malware categorization and detection. Every image in this dataset is associated with 25 malware types. A varied and comprehensive collection is offered for developing and evaluating of cybersecurity algorithms. The Malimg dataset's performance is evaluated using AlexNet, ResNet-18, and ResNet-50 – as shown in Table 12. The table shows findings for four nodes. These nodes are more likely to reflect different divisions of the Malimg dataset. Changes in the model's efficiency across nodes point to an unequal number of malware classes. If a model performs better on nodes having more instances of a given malware type, then it is efficient at classification. The right model is chosen based on its unique purpose. ResNet-50's high accuracy may make it preferable for decreasing false negatives. However, if achieving precise positive classifications. Then it is essential to avoid unwanted attacks. In this case, AlexNet may be the preferable option due to its higher precision. In the above table, ResNet-50 consistently gets the best accuracy over all four nodes varying from 92.86% to 94.57%. Whereas AlexNet has the lowest accuracy (73.43% to 94.57%). ResNet-18 and ResNet-50 have comparable F1 ratings across nodes. However, AlexNet has lower F1 values. When it comes to precision AlexNet has the best outcome from 92.00% to 94.57% for three of the four nodes. Whereas ResNet-50 has the least precision from 92.86% to 94.29% for three nodes. Fig 9 depicts a graph comparing the accuracy of several deep models between nodes. A deep model's average accuracy is 99.5%, while another's is 97.5%.

## 5.10. Comparison with test data

In malware detection tables are used to compare the outcomes of traditional and federated classification systems. Federated learning may be effective for privacy-sensitive or geographically distant datasets. Traditional training involves storing all of the data in a single location. The model is trained on the full dataset at once. Whereas FL involves distributing data across several devices or nodes. The model is trained on some part of the dataset on each device, and the changes are relayed to a central server, which updates the global model. FL framework it allows models to be learned on particular nodes. The efficacy of the federated learning model relative to traditional learning is determined by the dataset and model utilized. The accuracy of classification model training on two distinct datasets, Malimg [38] and Private Dataset is compared in Table 13 which presents the findings for AlexNet, ResNet-18, ResNet-50, and InceptionV3. Malimg has 25 classes, but Private Dataset has 28. The models' accuracy on the Malimg dataset is examined using both conventional and federated learning. Except for ResNet-50, all models perform better using traditional learning. For example, AlexNet gets 94.0% accuracy for traditional learning while achieving 93.80% with FL. On Private Dataset, FL outperforms all models except InceptionV3. For example, ResNet-50 reaches 96.70% accuracy on FL but just 96.33% with classical learning.

**Table 12. Performance metric of AlexNet, ResNet-18 and ResNet-50 on each node using Malimg dataset.**

| Node/Model | AlexNet | | | ResNet-18 | | | ResNet-50 | | |
|---|---|---|---|---|---|---|---|---|---|
| | Accuracy (%) | F-score | Precision (%) | Accuracy (%) | F-score | Precision (%) | Accuracy (%) | F-score | Precision (%) |
| Node 1 | 91.10 | 88.00 | 92.00 | 97.80 | 88.00 | 96.00 | 96.50 | 86.0 | 97.10 |
| Node 2 | 86.00 | 93.00 | 86.00 | 96.50 | 86.00 | 97.00 | 96.40 | 87.00 | 95.00 |
| Node 3 | 92.60 | 91.00 | 93.00 | 97.70 | 98.00 | 87.00 | 97.70 | 89.00 | 96.00 |
| Node 4 | 87.00 | 76.00 | 86.10 | 97.00 | 83.00 | 97.00 | 97.50 | 90.10 | 97.00 |

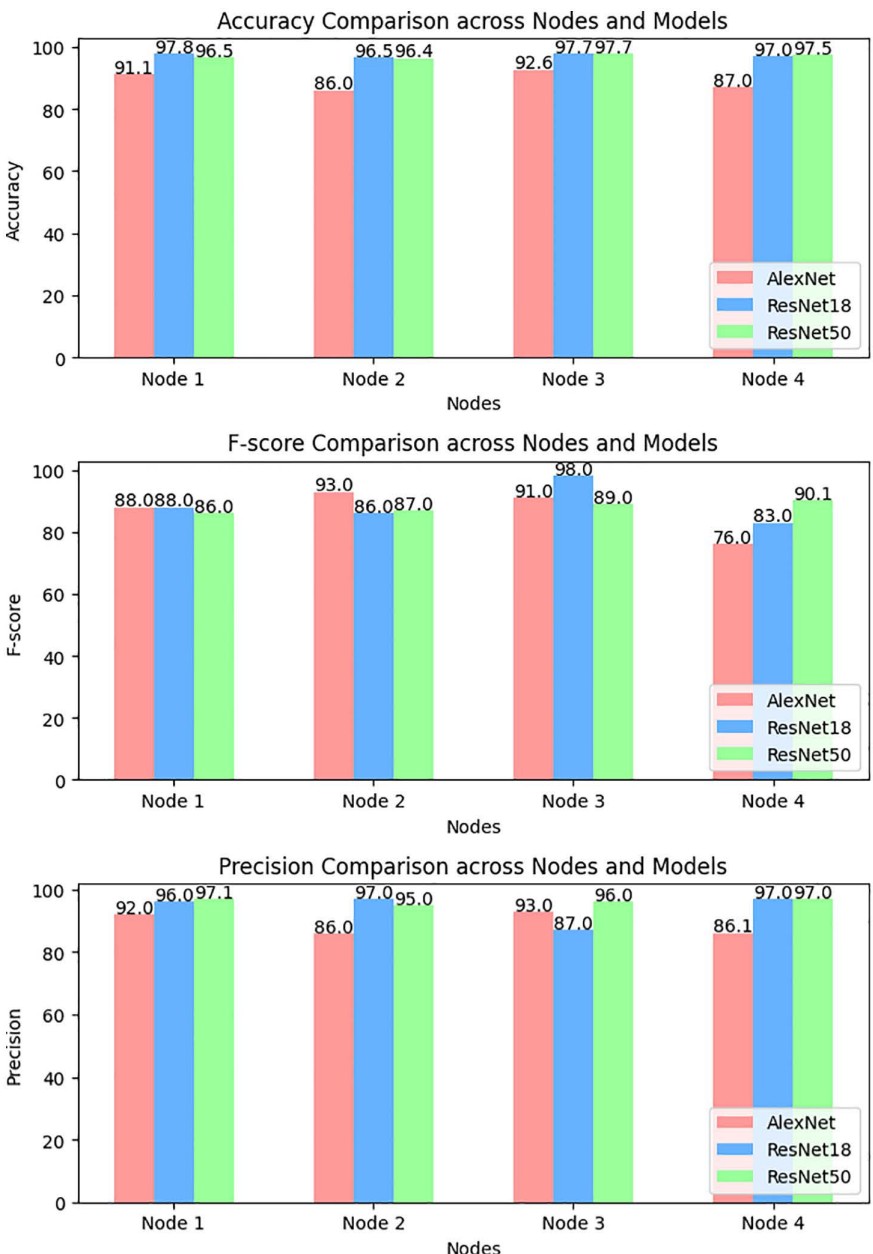

**Fig 9. Node level comparison of different deep models.**

## 6. Conclusion

To conclude, the challenge of classification and detection of the zero-day attacks in the cybersecurity persists because of huge number of Malware families and also the inherent imbalance data. Complexity highlights the pressing requirement for development of the strong defense method able of quickly recognizing these threats. Our study presents novel solution, Ensemble Federated Model, which is used to handle different range of malware kinds encompassing both similarities of inter and intra-class. Initially, our study adopts the malware Portable Executable (PE) files from internet sources and

**Table 13. Comparison of Malimg dataset and Private dataset.**

| Node/ Model | Malimg dataset (Accuracy) | | Private Dataset (Accuracy) | |
|---|---|---|---|---|
| | 25 classes | Average Ensemble Fed. (our) | 28 classes | Average Ensemble Fed. (our) |
| AlexNet | 94.0 | 96.12 | 93.80 | 90.90% |
| ResNet-18 | 94.7 | 95.34 | 93.89 | 95.41% |
| ResNet-50 | 93.60 | 96.33 | 96.70 | 97.80% |
| Inception-V3 | 93.1 | 76.00 | 96.80 | 98.20% |

thoroughly analyzes them by using three distinct antivirus programs through VirusTotal. Then, these files are changed into the image format and are classified into the 28 families. Employing transfer learning with a fine-tuned ResNet-50 model, the study extracts essential features from the images. These features, attained from various nodes in a distributed architecture, are then inserted into the novel Ensemble Stacked Federated Model for classification. The evaluation on both private and publicly available datasets determines the advantage of the proposed methodology over previous methods. By conducting individualized training on each node before merging them with a central model, ensuring enhanced performance and accuracy in malware classification.

## Supporting information

**S1 File. Supplementary material.**
(RAR)

## Author contributions

**Conceptualization:** Hassan Ishfaq.

**Data curation:** Hassan Ishfaq.

**Formal analysis:** Jamal Hussain Shah, Rabia Saleem, Maira Afzal.

**Investigation:** Maira Afzal.

**Methodology:** Hassan Ishfaq.

**Resources:** Maira Afzal.

**Supervision:** Jamal Hussain Shah.

**Validation:** Hassan Ishfaq, Rabia Saleem, Maira Afzal.

**Writing – original draft:** Hassan Ishfaq.

**Writing – review & editing:** Jamal Hussain Shah, Rabia Saleem, Maira Afzal.

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
