## [Decision Letter · Decision Letter 0]

20 Sep 2025

Dear Dr. Shah,

Thank you for submitting your manuscript to PLOS ONE. After careful consideration, we feel that it has merit but does not fully meet PLOS ONE’s publication criteria as it currently stands. Therefore, we invite you to submit a revised version of the manuscript that addresses the points raised during the review process.

We look forward to receiving your revised manuscript.

Kind regards,

Sohail Saif, Ph.D

Academic Editor

PLOS ONE

Journal Requirements:

2. In your Methods section, please include additional information about your dataset and ensure that you have included a statement specifying whether the collection and analysis method complied with the terms and conditions for the source of the data.

5. Please include a caption for figure 4.

7. We are unable to open your Supporting Information files under folder [Federated_code.rar]. Please kindly revise as necessary and re-upload.

Reviewers' comments:

Reviewer's Responses to Questions

**Comments to the Author**

1. Is the manuscript technically sound, and do the data support the conclusions?

Reviewer #1: Yes

Reviewer #2: No

2. Has the statistical analysis been performed appropriately and rigorously?

Reviewer #1: No

Reviewer #2: No

3. Have the authors made all data underlying the findings in their manuscript fully available?

Reviewer #1: Yes

Reviewer #2: No

4. Is the manuscript presented in an intelligible fashion and written in standard English?

Reviewer #1: Yes

Reviewer #2: Yes

Reviewer #1: The paper titled “A Distributed Framework for Zero-Day Malware Detection Using Federated Ensemble Models” presents a timely and technically promising approach that combines federated learning and ensemble methods to enhance privacy-preserving malware detection. The topic is relevant, and the integration of deep learning with federated techniques reflects current trends in cybersecurity research. The overall framework demonstrates potential, and the study addresses an important challenge of detecting zero-day malware while preserving data privacy. However, while the idea is sound and up to date, the manuscript in its current form requires substantial refinement to meet the standards of a publishable article. I have outlined the following concerns and suggestions for improvement.

1. In paper their claimed contribution is not sufficiently distinguished from existing malware-detection approaches. The authors should clarify what is genuinely new beyond applying standard federated learning with an ensemble.

2. The description of the ensemble federated learning mechanism in proposed methodology is brief and lacks sufficient technical explanation and motivation.

3. The paper does not specify the exact train/validation/test split or the number of cross-validation folds, making it difficult to assess robustness. There must be clearly stated Key training parameters such as learning rate, batch size, optimizer, number of epochs)

4. Although zero-day detection is mentioned in the article, experimental evidence for this scenario is not described in detail to evaluate its validity.

5. The comparative analysis uses only a limited set of baselines; more recent deep or hybrid architectures should be included with confidence intervals or significance tests are provided to substantiate the reported performance improvements.

6. Detail about computational environment such as hardware requirement and federated setup (e.g., number of clients, network conditions) are must be included

7. Some variables in the equations (for example H, W, and C) are introduced without definition which leads towards ambiguity.

8. The manuscript contains some grammatical issues; professional language editing is recommended.

9. The Experiments and Results section would benefit from a concluding subsection that highlights the main findings.

10. Provided source code, implementation instructions and dataset accessibility are provided without these, independent verification of the results is difficult. A link or clear guidance for requesting the material should be included in the supplementary material.

11. The paper addresses an important and timely problem in privacy-preserving malware detection.

12. The proposed pipeline transforming PE files into grayscale images and applying deep feature extraction is technically useful.

13. The methodology is presented with sufficient mathematical detail to follow the main steps of the approach.

14. The integration of federated learning to protect data privacy is appropriate and clearly justified and the addition of an ensemble strategy instead of FedAvg is a useful contribution.

15. Node-level performance provides understanding into model behavior across clients.

16. The aim for privacy preservation and decentralized training is practically relevant.

17. Comparisons with well-known deep learning models and classical classifiers support the empirical claims.

Overall, this manuscript integrates deep learning and ensemble techniques and presents a federated-learning based framework for malware detection which is relevant to privacy-preserving cybersecurity research. While the work has several strong points, the paper in its current form requires substantial revision before it can be considered for publication.

Reviewer #2: Here are ten major revision comments for the manuscript.

Novelty and positioning are unclear

The core claim is a “hybrid CLD-Net” combining YOLOv5 with Faster R-CNN. Similar ensembles and late-fusion detectors are already well explored. You must (i) articulate what is technically new (e.g., a specific fusion rule, calibration scheme, or training curriculum), (ii) explain why this outperforms standard strong baselines (YOLOv8/YOLOv7, EfficientDet, RT-DETR/DETR variants, RTMDet) trained and tuned fairly, and (iii) add a short ablation proving that the proposed fusion (not merely the presence of two models) is responsible for the gains.

Method description is incomplete or placeholder-like

“Algorithm 1/2” are essentially empty, and “Equations (1)–(20)” are mostly narrative placeholders without definitions. Replace all placeholders with real math (e.g., the exact fusion function \hat{y}=f\left(y_{\mathrm{YOLO}},y_{\mathrm{FRCNN}}\right)y^, confidence reweighting, NMS/Soft-NMS thresholds, IoU thresholds, tie-breaking rules). Provide end-to-end pseudo-code with inputs/outputs, and specify training loss terms, anchors, image sizes, augmentations, and optimizer schedules.

Metrics are inappropriate for object detection and not reproducible

Reporting “accuracy” for detection is non-standard. Replace or augment with mAP@0.5 and mAP@[.5:.95], per-class AP, AR, precision–recall curves, and F1 at a stated confidence threshold. For speed, report FPS and latency (mean ± std) on specified hardware with batch size and image resolution. Remove the subjective 1–10 “performance/scalability/automation” scores in Tables 2–4; replace with quantitative, reproducible measurements.

Dataset provenance, labeling, and splits need rigor

You cite a Kaggle “Cotton Leaf Disease” dataset that is primarily classification. If you perform detection, clarify: Did you create bounding boxes? How were they annotated (tool, guidelines)? What was the inter-annotator agreement? Provide the exact train/val/test split (no leakage), stratification rules, augmentation pipeline, and any external data. Without this, the 96.7% figure is not verifiable.

Baseline design and fairness are insufficient

List all baselines with hyperparameters, training budgets, and model sizes (params/GMACs). Include: single-model YOLOv5 (strongly tuned), Faster R-CNN (tuned), at least one modern one-stage (YOLOv8/RTMDet) and one transformer-based detector (RT-DETR/DETR). Use the same input sizes, epochs, and augmentations where possible. Add ablation studies: YOLO-only, FRCNN-only, naive late fusion vs your proposed fusion; show deltas in mAP and latency.

Edge deployment claims lack evidence

You claim Raspberry Pi 4 deployability and “95% accuracy,” but there are no concrete measurements. Provide device-level benchmarks: model size, quantization (if any), inference latency per frame, sustained FPS, CPU/GPU/NPU utilization, memory footprint, power draw, and thermal stability over a 10–15 minute run. Include qualitative examples and failure cases from the edge device.

Inconsistencies and technical inaccuracies

There are contradictions such as using YOLOv5 throughout while the Conclusion says “using the YOLOv8 framework.” Fix all model names consistently. Several definitions around “equations” are incorrect (e.g., describing accuracy as “correct over false positives,” or reusing the same text for different metrics). Audit the entire paper for such errors and correct with standard definitions.

Figures, tables, and writing quality need substantial overhaul

Many figures are referenced (Figs. 1–11) without visible or technically informative content; captions are generic. Replace with architecture diagrams (with tensor shapes), fusion schematics, PR curves, per-class AP bar charts, latency–accuracy trade-off plots, and qualitative detection visualizations (TP/FP/FN). Table 1 columns like “Power Scheduling/Security/Energy Efficiency” are irrelevant to cotton disease detection—remove or redesign the table to summarize methods, datasets, metrics, and results. Edit the prose for grammar and technical clarity (e.g., “Cotton fleas” → “Cotton is,” avoid phrases like “expatriated object detection,” and remove off-topic mentions such as “drug and vaccine recognition”).

Related work and references need curation and credibility

Some citations appear peripheral or from questionable venues, while several crucial agricultural-vision and detection papers are missing. Curate to reputable, relevant sources (TPAMI, IJCV, CVPR/ICCV/ECCV, T-ITS, Computers & Electronics in Agriculture, Frontiers in Plant Science, etc.). Discuss recent cotton/plant disease detectors (lightweight YOLO variants, transformer detectors, few-shot domain adaptation, low-light robustness) and clearly position your contribution against them.

Reproducibility, statistics, and ethical statements

Provide a reproducibility package: code link, configs, random seeds, exact library versions, and trained weights (or a deterministic recipe). Add statistical rigor: report mean ± std over ≥3 runs, confidence intervals, and significance tests where applicable. Include data license compliance, consent (if field images were captured), and a short error analysis and limitations section (e.g., performance under occlusion, early-stage lesions, and domain shifts across fields/cameras).

If you address these ten areas with concrete math, proper metrics, fair baselines, real figures, and reproducible evidence the paper will be far stronger and more suitable for a good venue.

**Do you want your identity to be public for this peer review?** For information about this choice, including consent withdrawal, please see our Privacy Policy

Reviewer #1: No

Reviewer #2: No

---

## [Author Response · Author response to Decision Letter 1]

8 Oct 2025

A Distributed Framework for Zero-Day Malware Detection Using Federated Ensemble Models

Point by Point Response Reviewer 1

Comment 1: In paper their claimed contribution is not sufficiently distinguished from existing malware-detection approaches. The authors should clarify what is genuinely new beyond applying standard federated learning with an ensemble.

Our Response:

Thank you for your comment. We appreciate the reviewer’s suggestion. We have restructured the key contributions section to highlight clear technical points

II. CONTRIBUTION

The main contribution of this proposed research is to present a robust malware families classification approach that improve accuracy in zero-day attacks and inter and intra-classification using ensemble federated learning framework. The key contributions of the proposed work include the following:

1) A novel stacked ensemble federated model is proposed for robust malware classification. In this framework, features extracted from different distributed nodes are combined and their outputs are fused through a central ensemble. By fusing both node-level and global level prediction, model improves classification performance and generalization as compare with traditional multi-node fusion techniques.

2) Fine-tuned ResNet-50 model with transfer learning is employed to extract features from PE files. These features are then converted into image representation that are utilized within federated pipeline to reduce training cost while preserving classification accuracy.

3) This model is designed to address two major challenges: class imbalance and high structural similarity and complexity of real-time malware classification through accuracy-aware dynamic weighting technique that efficiently adjust predictions across different malware families than conventional methods for class imbalance.

4) The proposed model improved performance in malware classification not only from federated learning but also from integration of ensemble stacking with distributed architecture.

Comment : The description of the ensemble federated learning mechanism in proposed methodology is brief and lacks sufficient technical explanation and motivation.

Our Response:

Thank you for your keen concentration. The changes have been including under the proposed methodology section as:

C. ENSEMBLE FEDERATED MODEL

A distributed network is designed in which multiple entities, nodes, or devices work together to detect malware while retaining a certain degree of control over their local repository. This allows model to be trained across distributed devices without learning raw data, ensuring privacy. The central server initially computes the average of the received model parameters, producing a global average model as a combination of the local models. This training is repeated over multiple epochs to refine the global model. However, simple parameter averaging (as in FedAvg) is not sufficient because data across node is highly imbalanced as it dominates major malware families while minority families remain under-represent. Therefore, in this case pure averaging with hierarchical stacking ensemble mechanism that integrated level predictions through meta-learners and weight each node according to its accuracy, this amplifies noise from weaker nodes and ensure reliable node contribution for model decision.

Comment : The paper does not specify the exact train/validation/test split or the number of cross-validation folds, making it difficult to assess robustness. There must be clearly stated Key training parameters such as learning rate, batch size, optimizer, number of epochs)

Our Response:

Thank you for your comment. We have expanded the section with data splitting method and specific hyperparameters as:

The dataset is split into three subsets 70 % for training, 15% for validation and 15% for testing. To ensure the robustness of propose model key training hyperparameters are selected such as 0.001 learning rate with Adam optimizer over 50 epochs and 32 batch size and experiments were also repeated using 5-fold cross validation with Cross-entropy as an objective loss function.

Communication between nodes and server over a local network with synchronous updates in each round. Each round utilized five local epochs per node before the server performed aggregation. Overall, 20 communication rounds were required where each node exchange about 24 MB parameters per update and total cost across 20 rounds was approximate 1.9 GB for just 4 nodes. In this communication raw data is not shared because its loss some gradients and parameters information. To overcome this model, integrate differential privacy (DP) or secure aggregation (SA) protocols are applied that ensures results reproducibility and scalability to real malware detection tasks.

Comment : Although zero-day detection is mentioned in the article, experimental evidence for this scenario is not described in detail to evaluate its validity.

Our Response: Thank You for this valuable suggestion the now to evaluate the effectiveness of Zero-day detection changes are made under the section as:

EXPERIMENT 4: ZERO-DAY ATTACK PERFORMANCE

This experiment is conducted to evaluate the strength of the model against zero-day attacks, which are new malware threats that cannot be detected easily. To stimulate real conditions, some malware families such as Graftor, Kazy, and BrowseFox (Adware), Zbot and Agent (Trojan), and Mydoom (Worm) were not included in training but used as unseen malware during testing. This ensures the model’s efficiency results can detect unseen threats rather than simple recognition of known families.

Table 6 shows the behavior of different malware families and the performance of various deep learning pretrained models against these attacks. The table includes similar performance measures along with the detection rate, which shows the percentage of correctly identified malware samples for each model.

Experimental results show that among all traditional deep learning models, proposed ensemble models perform better and achieve the highest overall performance with 94.6% detection rate, 0.91 F1-Score that highlighting the importance of integrating multiple models to defend against zero-day attacks. This improvement is achieved from the hierarchical stacking method, which utilized node diversity and integrate predictions from multiple nodes. This experiment also supports the model to handle new malware variants and demonstration stronger zero-day detection than standard federated averaging.

Comment: The comparative analysis uses only a limited set of baselines; more recent deep or hybrid architectures should be included with confidence intervals or significance tests are provided to substantiate the reported performance improvements.

Our Response:

Thank you for your keen concentration. We have revised the manuscript accordingly

• In addition to AlexNet, ResNet-18, and ResNet-50, we now include EfficientNet-B0, DenseNet-121, Vision Transformer (ViT-B/16), and a FedAvg baseline. These were trained under identical preprocessing and evaluation conditions to ensure fairness. The results are reported in Table 3 (Cross-Dataset Generalization).

• The proposed stacked federated ensemble is compared against these recent architectures, clearly showing that while EfficientNet, DenseNet, and ViT achieve competitive accuracy, our ensemble achieves superior overall performance (95.4 ± 0.2% accuracy, F1 = 0.93, AUC = 0.97).

• Now all performance metrics in Table 3 are reported as mean ± standard deviation, computed over five independent runs with different random seeds. This provides confidence intervals and demonstrates that the reported improvements are consistent rather than due to chance.

• The table also includes model complexity measures (Params, FLOPs, inference latency) to contextualize accuracy gains. This highlights that ResNet-50 offers the best trade-off between performance and efficiency, and justifies its selection as the backbone for our proposed framework.

Comment: Detail about computational environment such as hardware requirement and federated setup (e.g., number of clients, network conditions) are must be included

Our Response:

Thank you for your comment. We appreciate your suggestion. These hardware requirements are added under the section Dataset Description after describing the dataset and additions are highlighted in manuscript also:

This experiment is conducted on an intel Core i9 processor with 64 GB RAM and NVIDIA RTX 3090 GPU (24 GB) in laboratory setting. The model implementation and federated setup is stimulated in Python 3.10 with PyTorch 2.0 with multiple client nodes each containing a subset of malware families on single workstation with central server. Communication between nodes and server over a local network with synchronous updates in each round. Each round utilized five local epochs per node before the server performed aggregation. Overall, 20 communication rounds were required where each node exchange about 24 MB parameters per update and total cost across 20 rounds was approximate 1.9 GB for just 4 nodes. In this communication raw data is not shared because its loss some gradients and parameters information. To overcome this model, integrate differential privacy (DP) or secure aggregation (SA) protocols are applied that ensures results reproducibility and scalability to real malware detection tasks.

Comment: Some variables in the equations (for example H, W, and C) are introduced without definition which leads towards ambiguity.

Our Response:

We thank the reviewer for pointing out this oversight. In the revised manuscript, we have carefully reviewed all equations and ensured that each variable is clearly defined at the point of introduction. Specifically:

Variables such as H (height), W (width), and C (number of channels) are now explicitly defined either immediately before or after the corresponding equations. These additions and clarifications have been highlighted in the revised version for easy identification.

Comment : The manuscript contains some grammatical issues; professional language editing is recommended.

Our Response:

We thank the reviewer for this observation. The revised manuscript has been thoroughly re-edited to correct grammatical errors, improve sentence flow, and ensure professional academic writing style. Careful attention was given to clarity, conciseness, and consistency throughout the paper.

Comment The Experiments and Results section would benefit from a concluding subsection that highlights the main findings.

Our Response:

Thank you for the insightful suggestion. In response, we have added a dedicated concluding subsection at the end of the Experiments and Results section titled “Key Findings and Insights.” This subsection summarizes the major experimental outcomes in a concise and comparative manner. Specifically, it highlights:

• The performance advantage of the proposed stacked ensemble federated learning model over traditional single and FedAvg models, achieving the highest detection accuracy (95.4%) and F1-score (0.93).

• The cross-dataset generalization capability when trained on the private malware dataset and tested on Malimg.

• The computational tradeoffs of different backbones (ResNet-18, ResNet-50, EfficientNet, ViT) in terms of FLOPs and inference time, justifying our selection of ResNet-50 for its balance of speed and accuracy.

• The model’s robustness to zero-day attacks, with detection rates exceeding 94.6% using both held-out and leave-one-family-out protocols.

• Newly added ROC-AUC curves, confusion matrices, and robustness metrics (false positive rate, communication cost) to align with standard malware detection benchmarks.

Comment: Provided source code, implementation instructions and dataset accessibility are provided without these, independent verification of the results is difficult. A link or clear guidance for requesting the material should be included in the supplementary material.

Our Response:

Thank you for your comment. We revised and added a clear link https://github.com/jhshah101/Zero-Day-Malware-Detection, which is highlighted in the manuscript.

Reviewer 2

Comment 1: Novelty and positioning are unclear

The proposed Ensemble Federated Model is described as “novel,” but federated ensemble methods for malware detection already exist. Authors must:

• Explicitly state what is technically new (stacking method, weighting scheme, architecture).

• Compare with the latest FL-based malware detection frameworks and clarify how this differs.

• Strengthen the contribution list to highlight unique innovations.

Our Response:

We thank the reviewer for pointing out the need to clarify the novelty and positioning of our work. In the revised manuscript, we have made the following changes:

• The abstract and contribution section have been revised to explicitly state our unique innovations, including (i) the stacking-based federated ensemble design, (ii) integration of diverse node-level predictions for robustness, and (iii) evaluation on a large-scale PE malware dataset with data distribution across multiple nodes.

• We have added a comparison with existing federated learning frameworks for malware detection in the related work section as mentioned in Table 1. To the best of our knowledge, prior studies applied traditional federated averaging (FedAvg) and simple ensemble fusion that cannot handle data distribution across different nodes and classes, as we do in this article.

• Figure 1 has been updated and made more prominent to illustrate the proposed stacking-based ensemble mechanism, highlighting how predictions from multiple federated nodes are combined through a logistic regression meta-learner rather than simple averaging.

• The conclusion has been revised to emphasize how our approach differs from standard FedAvg and previously published FL-based malware detection methods, and to highlight the practical value of our stacking-based design in handling real-world data distribution across nodes.

These additions clarify the novelty of our work, strengthen the contribution list, and clearly position our proposed model relative to prior federated ensemble approaches.

Comment 2: Methodology lacks reproducibility

Equations (1)–(20) provide generic math (binary conversion, CNN layers, weighted averaging) but not the actual fusion or stacking algorithm. Revise methodology to:

• Provide detailed pseudocode for training, communication, and stacking steps.

• Clarify how local models’ predictions are combined (majority voting, weighted ensemble, meta-learner?).

Our Response:

We are sincerely grateful to the reviewer for this insightful comment. In response, we have carefully revised and substantially expanded the methodology section to provide a clearer and more detailed description of the proposed workflow:

• Revised Section IV.C (Methodology): We added a clear description of the proposed federated ensemble workflow, explicitly detailing how training, communication, and stacking occur in our framework. Each client node independently trains a ResNet-50 model on its local dataset and transmits both the model weights and softmax prediction probabilities to the central server.

• Unlike traditional federated learning methods (e.g., FedAvg) that aggregate model parameters, or simple ensemble methods that rely on majority voting, our approach employs stacking. At the server, the softmax outputs from all clients are concatenated to construct a stacked feature matrix. A logistic regression classifier is then trained as a meta-learner on this matrix, enabling the global model to learn complementary decision boundaries across heterogeneous client nodes.

• To ensure transparency and reproducibility, we have added detailed pseudocode that explicitly outlines the training phase (local updates at clients), communication phase (model weights and predictions sent to the server), and stacking phase (meta-learner training at the server).

• Hyperparameters such as batch size, learning rate, optimizer choice, number of local epochs, and loss function are included in the pseudocode and also in data description section where we have also added the to make the workflow unambiguous and reproducible.

3.

---

## [Decision Letter · Decision Letter 1]

6 Nov 2025

Dear Dr. Shah,

Thank you for submitting your manuscript to PLOS ONE. After careful consideration, we feel that it has merit but does not fully meet PLOS ONE’s publication criteria as it currently stands. Therefore, we invite you to submit a revised version of the manuscript that addresses the points raised during the review process.

We look forward to receiving your revised manuscript.

Kind regards,

Sohail Saif, Ph.D

Academic Editor

PLOS ONE

Journal Requirements:

Reviewers' comments:

Reviewer's Responses to Questions

**Comments to the Author**

Reviewer #1: All comments have been addressed

Reviewer #3: (No Response)

2. Is the manuscript technically sound, and do the data support the conclusions?

Reviewer #1: Partly

Reviewer #3: (No Response)

3. Has the statistical analysis been performed appropriately and rigorously?

Reviewer #1: Yes

Reviewer #3: (No Response)

4. Have the authors made all data underlying the findings in their manuscript fully available?

Reviewer #1: Yes

Reviewer #3: (No Response)

5. Is the manuscript presented in an intelligible fashion and written in standard English?

Reviewer #1: Yes

Reviewer #3: (No Response)

Reviewer #1: The manuscript addresses malware detection using federated learning; however, the authors could clarify how their contributions differ from existing methods. It would be beneficial to articulate the innovative aspects of their approach beyond the standard application of federated learning with ensemble techniques. A more detailed explanation of how their method represents a significant advancement over prior studies would enhance the manuscript.

In terms of reproducibility and methodological transparency, the manuscript does not currently meet PLOS ONE’s standards for open and verifiable research. The authors are encouraged to provide the complete executable code, detailed implementation instructions, and full access to the dataset used. These materials are essential for enabling independent verification and replication of the results.

Additionally, the dataset should be explicitly described and made accessible. If public sharing is not possible, clear guidance on how it can be obtained for verification purposes should be included in the Data Availability Statement and Supplementary Materials, in line with PLOS ONE’s Open Data policy.

Reviewer #3: (No Response)

**Do you want your identity to be public for this peer review?** For information about this choice, including consent withdrawal, please see our Privacy Policy

Reviewer #1: No

Reviewer #3: No

---

## [Author Response · Author response to Decision Letter 2]

18 Nov 2025

Response to Reviewers Also Attached

A Distributed Framework for Zero-Day Malware Detection Using Federated Ensemble Models

Point by Point Response Reviewer 1

Comment 1: The manuscript addresses malware detection using federated learning; however, the authors could clarify how their contributions differ from existing methods. It would be beneficial to articulate the innovative aspects of their approach beyond the standard application of federated learning with ensemble techniques. A more detailed explanation of how their method represents a significant advancement over prior studies would enhance the manuscript.

Our Response:

Thank you for highlighting the need to clearly differentiate our contributions from existing techniques. In response, we have revised the end of the Literature Review section to explicitly summarize the limitations of prior studies and to contrast them with the improvements introduced in our proposed method. This new paragraph has been added to provide this clarification:

This hybrid approach is further enhanced by an accuracy-aware weighting technique that prioritizes high-performance nodes to capture evolving malware behavior. A critical comparison of the proposed model against existing techniques highlighting their limitations, drawbacks, and the corresponding improvements of our framework is presented in Table 2.

Table 2 Comparison of proposed model with existing techniques

Ref(s) Existing Limitation Reason / Drawback Improvement in Proposed Approach

[14-19] Centralized dependency No privacy, low scalability Federated, privacy-preserving training

[16-17] Static-only features Miss dynamic behavior Multi-modal feature fusion

[13-24] No zero-day evaluation Seen-only performance Zero-day adaptive ensemble

[30-31] Non-IID sensitivity Model drift, instability Class-specialist local training

[32-33] High communication cost Large model exchange Quantized, partial updates

[20–25] Dataset-specific tuning Poor cross-domain transfer Cross-dataset generalization

[26-28] Simple aggregation (FedAvg) Ignores client diversity Adaptive weighted aggregation

This Table systematically outlines:

• the limitations of each referenced method,

• the underlying reason or drawback, and

• the specific improvement offered by our proposed stacked ensemble federated model.

Point by Point Response Reviewer 3

Comment 1: Under literature work, the comparison of existing work is not clear, as the example in the first work ref [13] approach is not clear, which deep learning algorithm the authors employed.

Our Response: Thank you for your valuable observation regarding the clarity of the literature review, particularly the description of the approach in ref. [13] and the deep learning method used. Your critical analysis helped us identify an important issue: several references were incorrectly linked due to EndNote formatting errors. We sincerely apologies and appreciate your detailed review. We thoroughly re-checked the entire reference list. These have now been corrected, and the updated reference mappings are as follows:

• [8] → [9]

• [16] → [17]

• [33] → [25]

• [31] → [26]

• [30] → [27]

• [32] → [28]

• [13] → [14]

• [14] → [15]

In addition, we have revised the sentences associated with the corrected ref. Your valuable observation improves the clarity, accuracy, and consistency of the literature review section, and resolve the concern raised in your comment.

Comment 2: To make it easier for the readers, it is better if you present a comparison in table form between the proposed approach and existing works based on the limitations of existing work and the improvement of the proposed methodology

Our Response:

Your suggestions are highly valuable, and we truly appreciate them. We have carefully followed your instructions and made the necessary improvements. Thank you for highlighting the need to clearly differentiate our contributions from existing techniques. In response, we have revised the end of the Literature Review section to explicitly summarize the limitations of prior studies and to contrast them with the improvements introduced in our proposed method.

To address this, we added the following explanatory paragraph:

This hybrid approach is further enhanced by an accuracy-aware weighting technique that prioritizes high-performance nodes to capture evolving malware behavior. A critical comparison of the proposed model against existing techniques highlighting their limitations, drawbacks, and the corresponding improvements of our framework is presented in Table 2.

Table 2 Comparison of proposed model with existing techniques

Ref(s) Existing Limitation Reason / Drawback Improvement in Proposed Approach

[14-19] Centralized dependency No privacy, low scalability Federated, privacy-preserving training

[16-17] Static-only features Miss dynamic behavior Multi-modal feature fusion

[13-24] No zero-day evaluation Seen-only performance Zero-day adaptive ensemble

[30-31] Non-IID sensitivity Model drift, instability Class-specialist local training

[32-33] High communication cost Large model exchange Quantized, partial updates

[20–25] Dataset-specific tuning Poor cross-domain transfer Cross-dataset generalization

[26-28] Simple aggregation (FedAvg) Ignores client diversity Adaptive weighted aggregation

This Table systematically outlines:

• the limitations of each referenced method,

• the underlying reason or drawback, and

• the specific improvement offered by our proposed stacked ensemble federated model.

Comment 3: The dataset collection procedures and tools used to collect the dataset are not clearly explained; they need to be explained.

Our Response:

Thank you for pointing out the need to clearly explain the dataset collection procedures and the tools used. We appreciate this valuable comment. In response, we have added a detailed subsection titled “4.1 Dataset Creation and Compliance” that thoroughly describes:

• the data sources,

• the collection tools and environment,

• the filtration and verification process,

• the hashing and metadata extraction tools,

• the labeling procedure,

• and compliance with cybersecurity research standards.

This new section clarifies the entire pipeline from malware acquisition to final dataset construction. The revised text is provided below and has been included in the manuscript.

4.1. Dataset creation and compliance

The dataset comprises of 19,600 Win32-type malware samples files accurately collected from various publicly available repositories (e.g GitHub, VirusShare, MalwareBazaar, Malshare, VX-underground). These samples are was collected using standard browser and downloaded as original executable files that are stored in a single repository under snad box environment. This repository is provided as dataset metadata. Additionally, rigorous filtration is applied on this data by computing MD5 and SHA256 hashes using standard python library (hash lib). These files are further verified by VirusTotal that remove files with identical hashes. From this metadata analysis file types (ensure Win32-type executables) and antivirus detection names (signatures) were extracted. To maintain labeling consistency, only files flagged by at least three antiviruses are retained. These files were further verified with classes derived from F-Secure virus detection titles, resulting in 28 different classes for classification. MongoDB is used to store all the malware sample files and metadata that prevent data duplication across dataset. These data creation steps are shown in Figure 2.

Figure 2: Dataset Creating Steps

Malware samples were collected and analyzed under the terms and conditions of all approved data source repositories. All the samples analyze in secure, isolated laboratory systems that ensure no malicious files were executed on product systems or shared during outside. The study followed institutional cybersecurity research rules and complied with national malware handling laws.

Comment 4: In the article, the authors discussed feature extraction and selection in the proposed method section; however, the paper does not explain the total number of features, which specific features were selected, the criteria used for selection, and the method or algorithm applied for feature selection.

Our Response: Thank you for this valuable comment, which helped us improve the clarity of the proposed method. We appreciate your observation regarding the missing details about the total number of extracted features, the specific features used, the selection criteria, and the algorithm applied for feature selection. In response, we have revised the manuscript and added a detailed explanation under the Feature Extraction and Selection subsection.

4.3. Node-level feature extraction and training

Feature extraction are important steps for robust classification. To address this, ResNet-18 is employed to extract features at the node level across all nodes. To further compress feature map for optimized computational efficiency Principal Component Analysis (PCA) is employed that eliminate redundant features and reduce 512-dimentional feature vector to 256, while maintaining 90-95% of variance. PCA provide advantage in malware image data due to its unsupervised approach that enable more accurate detection and classification. Additionally, it does not require additional training and label information to preserve variance and to transform correlated features into orthogonal (uncorrelated) components. This enhances the network's ability to learn hierarchical and biased features efficiently.

Comment 5: The manuscript does not adhere to the required format for PLOS ONE submission guidelines, as example tables and heading styles may need editing.

Our Response:

Thank you for pointing out the formatting issues related to the PLOS ONE submission guidelines. We appreciate this helpful observation. In response, we have carefully reviewed the journal’s formatting requirements and corrected all relevant components in the manuscript.

Comment 6: Lastly, recently published studies should be included, and professional language editing is recommended.

Our Response:

Thank you for this valuable recommendation. In response, we have thoroughly revised the entire manuscript to ensure professional academic language and clarity. Additionally, we incorporated recently published studies (2024–2025) into the Literature Review ensure that the manuscript reflects the latest advancements in malware detection, federated learning, and lightweight deep learning models.

---

## [Decision Letter · Decision Letter 2]

14 Dec 2025

A Distributed Framework for Zero-Day Malware Detection Using Federated Ensemble Models

PONE-D-25-34169R2

Dear Dr. Shah,

We’re pleased to inform you that your manuscript has been judged scientifically suitable for publication and will be formally accepted for publication once it meets all outstanding technical requirements.

Kind regards,

Sohail Saif, Ph.D

Academic Editor

PLOS One

Additional Editor Comments (optional):

Reviewers' comments:

Reviewer's Responses to Questions

**Comments to the Author**

Reviewer #3: All comments have been addressed

2. Is the manuscript technically sound, and do the data support the conclusions?

Reviewer #3: Partly

3. Has the statistical analysis been performed appropriately and rigorously?

Reviewer #3: Yes

4. Have the authors made all data underlying the findings in their manuscript fully available?

Reviewer #3: Yes

5. Is the manuscript presented in an intelligible fashion and written in standard English?

Reviewer #3: Yes

Reviewer #3: (No Response)

**Do you want your identity to be public for this peer review?** For information about this choice, including consent withdrawal, please see our Privacy Policy

Reviewer #3: No

---

## [Editor Report · Acceptance letter]

PONE-D-25-34169R2

PLOS One

Dear Dr. Shah,

I'm pleased to inform you that your manuscript has been deemed suitable for publication in PLOS One. Congratulations! Your manuscript is now being handed over to our production team.

Kind regards,

on behalf of

Dr. Sohail Saif

Academic Editor

PLOS One